# Remembrance of things practiced with fast and slow learning in cortical and subcortical pathways

James M. Murray [1,2✉] & G. Sean Escola [1,3]

The learning of motor skills unfolds over multiple timescales, with rapid initial gains in performance followed by a longer period in which the behavior becomes more refined, habitual, and automatized. While recent lesion and inactivation experiments have provided hints about how various brain areas might contribute to such learning, their precise roles and the neural mechanisms underlying them are not well understood. In this work, we propose neural- and circuit-level mechanisms by which motor cortex, thalamus, and striatum support motor learning. In this model, the combination of fast cortical learning and slow subcortical learning gives rise to a covert learning process through which control of behavior is gradually transferred from cortical to subcortical circuits, while protecting learned behaviors that are practiced repeatedly against overwriting by future learning. Together, these results point to a new computational role for thalamus in motor learning and, more broadly, provide a framework for understanding the neural basis of habit formation and the automatization of behavior through practice.

[1] Zuckerman Mind Brain and Behavior Institute, Columbia University, New York, NY 10027, USA. [2] Institute of Neuroscience, University of Oregon, Eugene, OR 97403, USA. [3] Department of Psychiatry, Columbia University, New York, NY 10032, USA. ✉email: jmurray9@uoregon.edu

The acquisition and retention of motor skills, for example shooting a basketball or playing a musical instrument, are crucial for humans and other animals. Such skills are acquired through repeated practice, and the effects of such practice are numerous and complex[1–3]. Most obviously, practice leads to improved performance, e.g., as measured by movement accuracy[4]. In addition, practice leads to the formation of habits, which can be defined operationally as decreased sensitivity of a behavior to goals and rewards[5,6]. Finally, practice leads to behavioral automatization, which can be measured by improved reaction times or decreased cognitive effort[2]. In many cases, these multiple effects of practice take place on different timescales, with large gains in performance obtained early in learning, followed by a much longer period in which the learned behavior becomes more refined, habitual, and automatic[7,8]. This suggests the possibility that motor learning can be divided into a fast, overt learning process leading to improved performance, combined with a slower, covert learning process with subtler effects related to habit formation and behavioral automatization. It is currently not well understood, however, whether these multiple learning processes that occur during practice are truly distinct or merely different facets of the same underlying process, whether they operate sequentially or in parallel, and how they might be implemented in the brain.

Neurobiologically, the descending pathway through the sensorimotor region of the striatum, the major input nucleus of the basal ganglia, plays an important role in both the learning and execution of learned motor behaviors[9–11], as well as in habit formation for highly practiced behaviors[12,13]. Such learning is facilitated by dopamine-dependent plasticity at input synapses to striatum, where this plasticity plays a role in reinforcing behaviors that have previously led to reward[14]. The sensorimotor striatum is able to influence behavior through two pathways: one descending pathway to motor-related brainstem nuclei[15] and another pathway to motor cortex via the basal ganglia output nuclei and motor thalamus[16]. Neural activity in sensorimotor striatum itself is driven by input from both cortical and subcortical sources. The dominant cortical input comes from sensorimotor cortex[17,18], and a great deal of research has implicated this pathway in motor learning[19,20]. The dominant subcortical input, meanwhile, comes from thalamus, including the rostral intralaminar nuclei, the centromedian nucleus, and the parafascicular nucleus[21–23]. Although this subcortical input to striatum has received less attention than the corticostriatal pathway, it has also been shown to represent behaviorally relevant variables[24–26] and to be important for the production of learned behaviors[27]. In general, however, the role of this thalamic pathway in motor learning and how it might differ from the corticostriatal pathway is not well understood.

In order to obtain some understanding about the roles that cortex, thalamus, and striatum play in motor learning, experimental manipulations of these structures and of the projections between them during different stages of learning provide useful constraints. Recent lesion and inactivation experiments in rats have shown that sensorimotor striatum and its thalamic inputs are necessary for both learning and execution of a skilled lever-press sequence[28]. In the same task, however, input from motor cortex to striatum is necessary for learning the task, but not for performing the task once it has already been learned[28,29]. Furthermore, the representation of task-related kinematic variables in striatum during the task is similar in rats with and without cortical lesions[30]. Evidence for the gradual disengagement of motor cortex during the learning of skilled movements has also been found in mice[31] and humans[32,33]. Together, the above results provide hints about the roles played by cortex, thalamus, and striatum in the acquisition and execution of skilled movements, suggesting that responsibility for driving learned behaviors may be gradually offloaded from cortical to subcortical circuits through practice. It is not immediately obvious, however, which neural- or circuit-level mechanisms might explain these results, nor how they might relate to the multiple effects of practice discussed above.

In this work, we propose a mechanistic theory of learning at cortico- and thalamostriatal synapses that provides a unified description of the lesion and inactivation results described above and, more broadly, points to computationally distinct roles for the two inputs to striatum in the acquisition and retention of motor skills. As a starting point for addressing learning, retention, and recall accuracy, we begin by modeling a single neuron within striatum. We mathematically derive the neuron's forgetting curve, which quantifies the probability that the neuron produces the correct output in response to a particular pattern of inputs as a function of how far in the past learning occurred. If the neuron's input synapses are all modified by a supervised (i.e., error-minimizing) plasticity rule, then the neuron learns to produce the correct output in response to a particular input, but there is no added benefit to repeated practice once the output has been learned. If, on the other hand, some of the neuron's input synapses are modified by a slower, associative learning rule, then patterns that are practiced many times become far more resistant to being overwritten by future learning. We interpret the first input, with fast supervised learning, as motor cortex, and the second input, with slow associative learning, as thalamus. We further show that, as a behavior is practiced, there is a gradual and covert transfer of control, with the activity of the downstream population increasingly driven by the second input pathway. In addition to providing a unified description of the lesion and inactivation experiments described above, this model generates predictions for the expected effects of perturbations to the cortical and thalamic pathways. Further, it proposes a new computational role for thalamic inputs to striatum and, more broadly, provides a framework for understanding habit formation and the automatization of behavior as the control of behavior is transferred from cortical to subcortical circuits.

## Results

In the following sections, our aim will be to develop a theory that describes the effects of learning and practiced repetition described above. As a first step, we will mathematically investigate learning and forgetting within a simple model of supervised learning in a single striatal neuron receiving cortical inputs. This will provide a minimal quantitative theory for addressing questions about retention and accuracy in learned behaviors, but will not yet account for effects related to practiced repetition. In order to account for such effects as well as for the lesion and inactivation results described above, we will next add a second set of inputs from thalamus. By endowing these inputs with associative learning, we will show that this second input pathway leads to enhanced retention and greater accuracy for patterns that are practiced many times during training. These two types of learning together give rise to a covert learning process in which control of the striatal population is gradually transferred from the cortical to the thalamic inputs through repeated practice. Finally, we describe how this two-pathway model leads to a number of experimental predictions about the expected effects of perturbations and inactivations of the cortical and thalamic inputs.

**A single-neuron model describes learning and forgetting**. In order to develop a quantitative theory of learning at the most basic level, we began by studying a classical model of a single neuron: the perceptron[34], which in our model corresponds to a single striatal neuron receiving cortical input (Fig. 1a, b). In this model, $N_x$ input signals $x_i$ are multiplied by synaptic weights $w_i$ and then summed. If the summed value is greater than 0, the

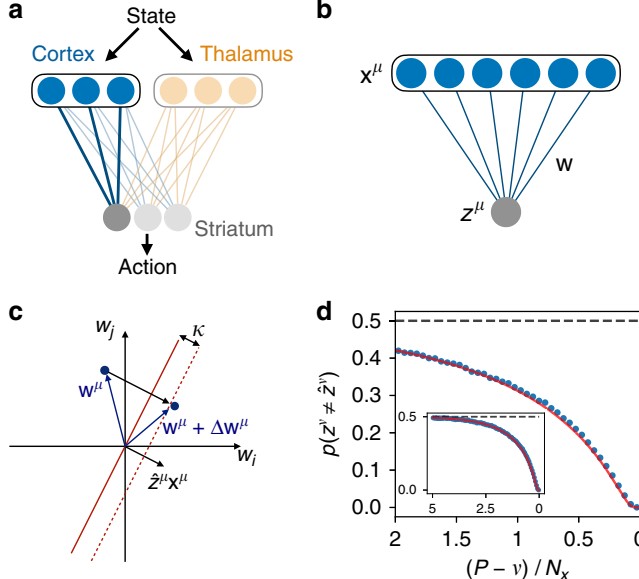

**Fig. 1 The forgetting curve for a single neuron with supervised learning.**
**a** A sensorimotor circuit model in which cortex and thalamus contain state information (e.g., cues and sensory feedback) and provide input to striatum, which learns to drive actions that lead to reward. **b** A model of a single striatal neuron receiving cortical inputs, where weights **w** are trained such that random input patterns $\mathbf{x}^\mu$ produce the correct classifications $z^\mu = \hat{z}^\mu$. **c** The input vector $\mathbf{x}^\mu$ defines a hyperplane (red line) in the space of weights **w**, and the update rule modifies the weight vector $\mathbf{w}^\mu$ to give the correct classification of pattern $\mu$ with margin $\kappa = 1$ (dashed red line). **d** The probability of incorrect classification when testing pattern $\nu$ after learning $P = 2N_x$ patterns sequentially. The most recently learned patterns are on the right, with earlier patterns on the left. The solid curve is the theoretical result; points are simulated results. Inset shows the same result for a perceptron trained with $P = 5N_x$ patterns.

neuron's output is $+1$, otherwise the output is $-1$. This is described by the equation $z^\mu = \mathrm{sgn}(\mathbf{w} \cdot \mathbf{x}^\mu)$, where $\mu$ denotes one of $P$ possible input patterns (Fig. 1b). The goal is then for the synaptic weights **w** to be adjusted so that, for each pattern $\mu$, the output matches a target value $\hat{z}^\mu$. Thus, the task is binary classification, in which random input patterns $x_i^\mu \sim \mathcal{N}(0,1)$, where $\mathcal{N}$ denotes the standard normal distribution, are mapped onto random output values $\hat{z}^\mu = \pm 1$. In our neurobiological interpretation, the perceptron corresponds to a single neuron in striatum, the major input structure of the basal ganglia. This neuron receives inputs from cortex with information that may represent cues, context, and sensory feedback, and it learns to adjust the strengths of its input weights so that the neuron is active in response to some inputs and inactive in response to others.

This learning problem can be illustrated geometrically in the $N_x$-dimensional space of synaptic weights, where the weights just before learning pattern $\mu$ are represented as a vector $\mathbf{w}^\mu$ in this space (Fig. 1c). In order for the classification to be correct, the weight vector after training should have positive overlap with the vector $\hat{z}^\mu \mathbf{x}^\mu$, which defines a classification boundary (red line in Fig. 1c), with all weight vectors on one side of this boundary leading to correct classification. This can be effected by updating the weights to $\mathbf{w}^\mu + \Delta \mathbf{w}^\mu$, where

$$\Delta \mathbf{w}^\mu = \begin{cases} (\kappa \hat{z}^\mu - u^\mu)\mathbf{x}^\mu / N_x, & u^\mu \hat{z}^\mu < \kappa, \\ 0, & \text{else,} \end{cases} \quad (1)$$

where $u^\mu = \mathbf{w}^\mu \cdot \mathbf{x}^\mu$ is the summed input to the neuron[35]. The first line of this equation says that, if the classification is initially

incorrect or correct but within a margin $\kappa$ of the classification boundary (dashed line in Fig. 1b), then an update is made such that the new weight vector $\mathbf{w}^\mu + \Delta \mathbf{w}^\mu$ lies on the correct side of the classification boundary (Fig. 1c) with margin $\kappa$. The second line, on the other hand, says that, if the classification is initially correct with a margin of at least $\kappa$, then no update needs to be made. Because $\kappa$ can be absorbed into an overall rescaling of **w**, we henceforth set $\kappa = 1$.

Clearly a potential problem exists whenever more than one pattern is being learned, however. After applying (1) to ensure correct classification of pattern $\mu$, followed by applying the same update rule for pattern $\mu + 1$, there is no guarantee that the weight vector will still be on the correct side of the classification boundary for pattern $\mu$. That is, the learning of new information in this model has the potential to interfere with and overwrite previously learned information. In the standard paradigm for learning in the perceptron, this problem is solved by cycling many times through all of the $P$ patterns to be learned. In one of the most celebrated results in computational neuroscience, it has been shown that, in the limit of large $N_x$, a weight vector producing correct classification of all patterns can be obtained whenever $P < 2N_x$[36,37].

In order to address effects related to retention and recall, we instead study the case in which the patterns are learned in sequence, so that no further training with pattern $\mu$ occurs after training with pattern $\mu + 1$ has begun. Concretely, we first train the perceptron by applying the update rule (1) for each of the $P$ patterns in the sequence. We then test the perceptron by testing the classification of each pattern using the final weight vector $\mathbf{w}^P$. Intuitively, we expect that the most recently learned classifications will remain correct with high probability during testing since the weight vector will have been updated only a small number of times after learning these patterns. On the other hand, for classifications learned much earlier and hence overwritten by a large number of subsequent patterns, the probability of incorrect classification should approach chance level.

In order to describe this process of training, overwriting, and testing quantitatively, we calculated the probability of incorrect classification (i.e., the error rate) for each pattern during testing after all patterns have been trained. The full calculation is presented in Supplementary Note 1. To summarize, it begins by assuming the weight vector is $\mathbf{w}^\nu$ just before learning some generic pattern $\nu$, then applies the update rule (1) using randomly drawn $\mathbf{x}^\nu$ and $\hat{z}^\nu$ to obtain $\Delta \mathbf{w}^\nu$. For the following $P - \nu$ steps ($\mu = \nu + 1, \ldots, P$), the updates $\Delta \mathbf{w}^\mu$ will be in directions that are random with respect to $\mathbf{w}^\nu$. Hence, the evolution of the weight vector from $\mathbf{w}^\nu$ to $\mathbf{w}^P$ can be described as a drift-diffusion process characterized by the probability distribution $p(\mathbf{w}^P | \mathbf{w}^\nu + \Delta \mathbf{w}^\nu)$. Once the probability distribution for $\mathbf{w}^P$ is known, the error rate during testing is calculated by averaging $z^\nu = \mathrm{sgn}(\mathbf{w}^P \cdot \mathbf{x}^\nu)$ over $\mathbf{w}^P$, as well as over the initial weight vector $\mathbf{w}^\nu$ and the random input pattern $\mathbf{x}^\nu$.

The result of this calculation is that the error rate during testing of pattern $\nu$ (for clarity, we denote patterns by an index $\mu$ during training and by an index $\nu$ during testing) is given by

$$p(z^\nu \neq \hat{z}^\nu) = F\left(\frac{P - \nu}{N_x}\right), \quad (2)$$

where the form of the function $F$ is given in Supplementary Note 1. This result defines the single-neuron forgetting curve, which is shown in Fig. 1d. (While the pattern index $\nu$ is the independent variable in the forgetting curve, we plot the curve as a function of $(P - \nu)/N_x$ so that earlier times appear on the left and later times appear on the right, and so that the curve is independent of $P$, which may be arbitrarily large.) As expected, the forgetting

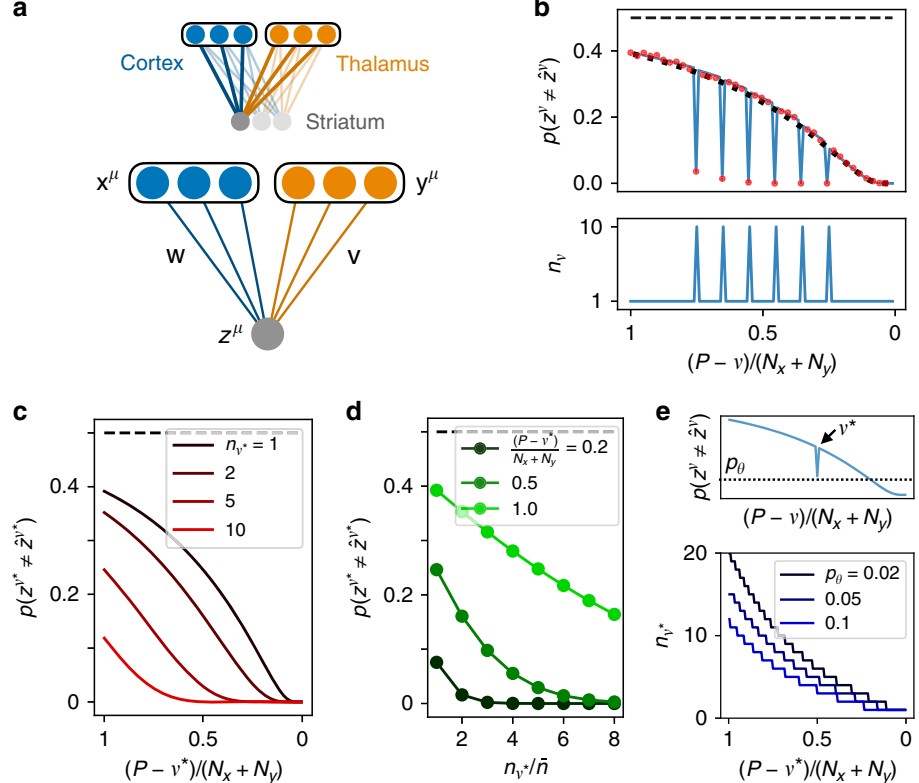

**Fig. 2 Repetition during training improves recall in the two-pathway model. a** The two-pathway architecture, with cortical and thalamic inputs to striatum (top), where fast supervised learning of corticostriatal weights **w** is accompanied by slow Hebbian learning of thalamostriatal weights **v** (bottom). **b** The forgetting curve (top) in a case where six patterns are repeated multiple times during training, while all other patterns are presented only once (bottom). Solid curve is theoretical result ($\alpha = 1$, $\beta = 1$); points are simulations with $N_x = N_y = 1000$; dotted curve shows the case in which no patterns are repeated multiple times. **c** The error rate for pattern $\nu^\star$, which is repeated $n_{\nu^\star}$ times during training, while other patterns are presented only once. **d** The error rate as a function of $n_{\nu^\star}$, with curves corresponding to different choices of $\nu^\star$. **e** Top: A single pattern $\nu^\star$ is trained multiple times while all other patterns are trained once. Bottom: the number of times that pattern $\nu^\star$ must be repeated during training in order to obtain a classification error rate below a threshold $p_\theta$ during testing.

curve in Fig. 1d shows that recently learned classifications are recalled with very low error rates, while those learned far in the past are recalled with error rates approaching chance level. We can further observe that, because the number of remembered patterns and the number on inputs appear only through the ratio $(P - \nu)/N_x$, the number of patterns that can be learned with error rate at or below any particular threshold is proportional to $N_x$. This means that the memory is extensive in the number of synapses, as in the classical result described above for the perceptron with cycled training. Finally, because this model implements learning in a single step and may therefore be questionable as a neurobiological model of learning, we show using simulations that a similar forgetting curve is obtained using a gradient-descent learning rule, in which small updates are accumulated over many repetitions to minimize the readout error (Supplementary Fig. 1).

The single-neuron model that we have studied in this section provides a zero-parameter baseline theory for addressing questions related to learning, retention, and recall accuracy. The ability of this model to learn new information by overwriting recently learned information is in contrast to the case of the classical perceptron, in which patterns are cycled through repeatedly during training, leading to a classification error rate of zero for up to $P = 2N_x$ patterns. However, no further classifications could be learned in this case without drastically impairing the classification performance of the already-learned patterns. This phenomenon, known as catastrophic forgetting[38], is obviated in the sequentially trained model studied here. As previously pointed out in theoretical studies

of related models[39,40], although sequentially trained neurons and neural networks have a smaller overall memory capacity than models in which training is repeated, they have the advantage that learned patterns decay smoothly over time as new patterns are learned, even as $P/N_x \to \infty$.

**A two-pathway model accounts for the effects of practice.** In the preceding section, we derived the forgetting curve for a single-neuron model of a striatal neuron receiving cortical inputs with supervised learning in order to quantitatively address learning and forgetting. Because learning of any given pattern occurs in a single step and stops once the error is zero, however, this simple model is unable to describe the effects of practice. In order to study the effects of practice, we made the simplest possible addition to the model by including a second input pathway, where synaptic weights in this pathway are modified with an associative update rule, as originally proposed by Hebb[41] (Fig. 2a). Specifically, these synapses are strengthened by coactivation of their inputs and outputs, so that the same inputs will in the future tend to produce the same outputs. This strengthening, which is independent of errors or rewards, continues to accumulate with each repetition if an input pattern is presented many times. Importantly, the Hebbian learning should be slow compared to the supervised learning, so that it will not strengthen an incorrect input–output association before the supervised learning has taken place. Intuitively, then, this picture suggests that repeated training of a particular classification should strengthen

the association and make it more resistant to being overwritten by later learning. We shall show below that this is indeed the case. After developing the theory for this two-pathway model, we shall show in the following section that identifying the second input to striatum as thalamus provides an explanation for the asymmetric effects of motor cortical and thalamic lesions in rats described in the "Introduction"[28–30].

Concretely, we extended the single-neuron model by adding a second set of inputs $\mathbf{y}^\mu$, where the $N_y$ inputs $y_i^\mu \sim \mathcal{N}(0,1)$ are random and independent of the first set of $N_x$ inputs $x_i^\mu$. The output is then given by $z^\mu = \mathrm{sgn}(\mathbf{w} \cdot \mathbf{x}^\mu + \mathbf{v} \cdot \mathbf{y}^\mu)$, where the synaptic weights $\mathbf{v}$ are updated according to a Hebbian learning rule. Specifically, if $\mathbf{v}^\mu$ is the weight vector just before training on pattern $\mu$, then the weight vector just after training on pattern $\mu$ is given by $\mathbf{v}^\mu + \Delta\mathbf{v}^\mu$, where

$$\Delta\mathbf{v}^\mu = -\frac{\alpha n_\mu}{N_y \bar{n}} \mathbf{v}^\mu + \sqrt{2} \frac{\beta n_\mu}{N_y \bar{n}} \hat{z}^\mu \mathbf{y}^\mu. \tag{3}$$

The second term in this equation is a modification to the synaptic weight that is proportional to the product of the pre- and post-synaptic activity, making this a Hebbian learning rule. Because the learning of $\mathbf{w}$ is assumed to be fast relative to this Hebbian learning, the downstream activity will have attained its target value by the time the Hebbian learning has any significant effect, allowing us to use $\hat{z}^\mu$ rather than $z^\mu$ in Eq. (3). The $n_\mu$ appearing in Eq. (3) is interpreted as the number of times that the pattern $\mu$ is repeated when it is trained, and $\bar{n}$ is the average of $n_\mu$ over all patterns.

The first term in Eq. (3) causes the weights to decrease slightly in magnitude with each repetition of each pattern, so that they do not become arbitrarily large after a large number of classifications have been learned. The constants $\alpha$ and $\beta$ control how quickly the weights decay and how strong the initial updates are, respectively. The weights $\mathbf{w}$, meanwhile, are updated as before, except that now the total summed input appearing in Eq. (1) is given by $u^\mu = \mathbf{w} \cdot \mathbf{x}^\mu + \mathbf{v} \cdot \mathbf{y}^\mu$. While the prefactors $N_y$ and $\bar{n}$ can clearly be absorbed into a redefinition of the learning and forgetting rates and are hence arbitrary from a biological point of view, explicitly including them simplifies the mathematical expressions that follow from Eq. (3) and leads to a theory with a sensible $N_y \to \infty$ limit.

In our neurobiological interpretation, the second pathway corresponds to thalamic inputs to the striatal neuron. As suggested by experiments[24–26], these inputs, like cortical inputs, may convey task-relevant information about cues and sensory feedback. Unlike the cortical inputs, however, the thalamic inputs connect to the readout neuron with synapses that are modified by a Hebbian update rule.

For this two-pathway model, we derived a generalized forgetting curve using the same approach as that used for the perceptron, describing the evolution of both weight vectors $\mathbf{w}$ and $\mathbf{v}$ as drift-diffusion processes and computing statistical averages over weights and patterns. The generalized forgetting curve for the two-pathway model has the form

$$p(z^\nu \neq \hat{z}^\nu) = G\left(\frac{P-\nu}{N_x}, \frac{N_y}{N_x}, \alpha, \beta, \frac{n_\nu}{\bar{n}}\right), \tag{4}$$

where the function $G$ is derived in Supplementary Note 2. From Eq. (4), we see that the forgetting curve now additionally depends on the Hebbian learning rate $\beta$, the ratio $N_y/N_x$, the Hebbian decay rate $\alpha$, and the number of repetitions $n_\nu$ for each pattern. In Supplementary Figs. 2 and 3, we fixed $n_\nu$ to be constant for all patterns and investigated the effects of $\alpha$, $\beta$, and $N_y/N_x$ on the forgetting curves. We found that, while adding the second pathway does not in general improve the memory performance

per synapse (of which there are $N_x + N_y$), it does improve the memory performance per supervised synapse (of which there are $N_x$). Further, when the Hebbian decay rate $\alpha$ becomes small, the number of very old patterns that can be correctly recalled is improved, even if normalizing by the total number of input synapses (i.e., by $N_x + N_y$, rather than by $N_x$).

We next investigated the case in which some patterns are repeated more than others during training. In this case, we found that repeating particular patterns consecutively during training causes those patterns to be retained with much lower error rates and for much longer than nonrepeated patterns. If trained with enough repetitions, particular patterns can be recalled essentially perfectly even after $\sim N_x + N_y$ additional patterns have been learned (Fig. 2b).

Further, as long as the number of highly practiced patterns is much less than $N_x + N_y$, the error rate for the remaining patterns is not significantly increased (solid line versus dotted line in Fig. 2b). This shows that it is possible for a neuron to become an "expert" at a small number of classifications without performing significantly worse on other classifications that have recently been learned. In Supplementary Fig. 4, we show that this remains true as long as the number of repeated patterns is much smaller than the total number of inputs $N_x + N_y$.

The underlying reason for the selectively enhanced recall of practiced patterns is the reinforcement of the association between the input pattern and the target output value via the updates to the synaptic weights in the second pathway, where this association is strengthened through repetition. As we shall show in the next section, this process can be understood as a transfer of control from the first to the second pathway as a pattern is repeated multiple times.

We next investigated the dependence of the error rate for a particular pattern $\nu^*$ on the interval (the "testing interval") between training and testing, as well as on the number of repetitions $n_{\nu^*}$, for that pattern, again assuming that all other patterns are presented only once during training. We found that the effect of practice is to shift the forgetting curves in a roughly parallel manner (Fig. 2c). This parallel shifting of forgetting curves via practiced repetition is a well-known result from experimental psychology studies in human subjects performing memory tasks[42–46]. Plotting the data instead as a function of the number of repetitions $n_{\nu^*}$ for a fixed testing interval shows the error rate smoothly decreasing with practice (Fig. 2d), again bearing similarity to results from memory studies in humans[47]. This shows that the error rate during testing, for a fixed testing interval, decreases as a function of how many times that pattern was practiced.

We next asked, if just a single pattern $\nu^*$ is repeated multiple times while all other patterns are trained just once, how many times must pattern $\nu^*$ be repeated in order to obtain an error rate below a threshold $p_\theta$ during later testing of that pattern? The number of necessary repetitions was found to be a supralinear function of the interval between the training and testing of pattern $\nu^*$ (Fig. 2e). This dependence could potentially be (and, to our knowledge, has not already been) measured experimentally, for example in human subjects learning paired associations (as in, e.g., refs. [45,46]), with certain associations presented multiple times in succession during training, and with varying intervals between training on these repeated patterns and testing.

## Input alignment and control transfer in the two-pathway model

In the previous section, we showed that, due to the reinforcement of the input-to-output mapping that comes about by slow Hebbian learning in the single-neuron model, input from the second pathway can contribute to driving the readout unit to

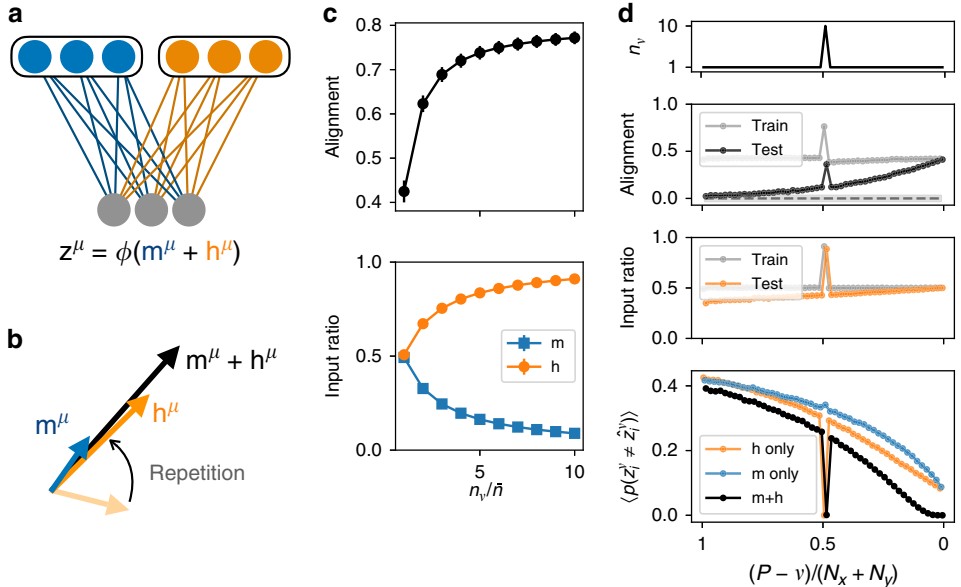

**Fig. 3 Inputs to the downstream population become aligned through repetition. a** A downstream population receives input from two pathways, where the first input **m** is trained with fast supervised learning, while the second input **h** is modified with slow Hebbian learning. **b** Through repetition, the second input becomes increasingly aligned with the first, while its projection along the readout direction $\hat{\mathbf{z}}$ grows. **c** Top: The normalized overlap between **m** and **h** for one particular pattern that is repeated $n_\nu$ times. Bottom: The ratio of the projection of each input component along the readout direction ($\mathbf{m} \cdot \hat{\mathbf{z}}$ or $\mathbf{h} \cdot \hat{\mathbf{z}}$) to the total input along the readout direction ($(\mathbf{m} + \mathbf{h}) \cdot \hat{\mathbf{z}}$). **d** Top: In a two-pathway network trained on $P$ patterns sequentially, one pattern is repeated 10 times during training, while others are trained only once. Middle panels: The alignment for each pattern during sequential training and testing (upper panel, where shaded region is chance alignment for randomly oriented vectors), and the ratio of the input **h** along $\hat{\mathbf{z}}$ to the total input along $\hat{\mathbf{z}}$ (lower panel). Bottom: The error rate, averaged over $N_z$ readout units, for the trained network with either both inputs intact (black) or with one input removed (blue and orange) (curves offset slightly for clarity). All points in (**c**) and (**d**) are simulations with $\alpha = \beta = 1$ and $N_x = N_y = N_z = 1000$, averaged over $n = 1000$ randomly initialized networks. Error bars denote standard deviations and are, where not visible, smaller than the plotted points.

its target state. We next investigated how the learning process in the two-pathway model could be interpreted at the population level. In this case, while each individual readout neuron performs the same binary classification task as before, the activity of the readout population is characterized by an $N_z$-dimensional vector **z**. Interpreting the two input populations as cortical and thalamic inputs to striatum, we asked what are the principles that determine the relationship of this striatal population activity to its inputs, and how does this relationship evolve through learning. To address this, we defined **m** and **h** as the $N_z$-dimensional inputs to the readout population from the first and second pathway, respectively (Fig. 3a), where $m_i = \sum_{j=1}^{N_x} W_{ij} x_j$ and $h_i = \sum_{j=1}^{N_y} V_{ij} y_j$, and the weight vectors **w** and **v** from the single-neuron model have been promoted to weight matrices **W** and **V**. The activity in the readout population is then described by the $N_z$-dimensional vector $\mathbf{z} = \text{sgn}(\mathbf{m} + \mathbf{h})$. In this case, applying the updates (1) and (3) for each synapse when training on pattern $\mu$ leads to an updated input current from the first pathway of $\mathbf{m}^\mu + \Delta \mathbf{m}^\mu = \hat{\mathbf{z}}^\mu - \mathbf{h}^\mu$, where we have used $\Delta \mathbf{m}^\mu = \Delta \mathbf{w}^\mu \cdot \mathbf{x}^\mu = \hat{\mathbf{z}}^\mu - \mathbf{m}^\mu - \mathbf{h}^\mu$. Similarly, the input current from the second pathway becomes $\mathbf{h}^\mu + \Delta \mathbf{h}^\mu = [1 - \alpha n_\mu/(\bar{n} N_y)] \mathbf{h}^\mu + \sqrt{2}\beta n_\mu \hat{\mathbf{z}}^\mu/\bar{n}$. Thus, following the updates, both of the input currents obtain a component of magnitude $\sim O(1)$ along the target activity $\hat{\mathbf{z}}^\mu$, so that they become aligned with the target activity and with each other. Further, for the Hebbian update, this component continues to grow as the pattern is repeated $n_\mu$ times in succession (illustrated in Fig. 3b).

As a single input pattern is repeated multiple times, the normalized alignment between the inputs, defined as $\mathbf{m} \cdot \mathbf{h}/(|\mathbf{m}||\mathbf{h}|)$, grows. As described above, this increased alignment occurs because **h** becomes increasingly aligned with $\hat{\mathbf{z}}$, and hence

also with **m**, as the number of repetitions increases. This means that the inputs from the two pathways are driving the downstream population in similar ways (Fig. 3c, top), a process that we refer to as input alignment.

In addition to this process of input alignment, the input from the second pathway along $\hat{\mathbf{z}}$ constitutes an increasingly dominant proportion of the total input along $\hat{\mathbf{z}}$ as the pattern is repeated (Fig. 3c, bottom). Thus, the downstream activity is driven relatively less by the first input and more by the second input as a pattern is repeated. We refer to this process as control transfer. Both of these processes—alignment of the two inputs and the transfer of control from one pathway to the other—are covert, in the sense that they occur gradually during repetition and do not have any immediately obvious behavioral correlate, since the learning in the first pathway causes the activity in the downstream population to attain its target value already by the first repetition.

In order to further illustrate the implications of input alignment and control transfer, we sequentially trained a two-pathway network with a multi-neuron readout population to produce $P$ classifications, generalizing the task from previous sections to the case $N_z > 1$. During training, one particular pattern was repeated many times in succession, while all others were trained only once (Fig. 3d, top). We found that input alignment became large and positive during training and, particularly for the repeated pattern, remained large and positive during testing after training was complete (Fig. 3d, second panel). Similarly, the ratio of the input **h** along $\hat{\mathbf{z}}$ from the second pathway to the total input along $\hat{\mathbf{z}}$ became large for the repeated pattern during training and remained large during testing, illustrating the occurrence of control transfer (Fig. 3d, third panel). Finally, when testing the error rate for each pattern after training was complete, we

performed "lesions" by shutting off each of the inputs during testing (Fig. 3d, bottom). In the case $\mathbf{m} = 0$, we observed that the error rate increased for all patterns except for the pattern that was repeated multiple times during training, which retained an error rate near zero. This shows that, via the mechanisms of input alignment and control transfer for the highly practiced behavior, the second pathway was able to compensate for the loss of the first pathway, driving the downstream population to produce the correct activity for this pattern even in the absence of the first input. In the case $\mathbf{h} = 0$, on the other hand, the repeated pattern was not recalled accurately, illustrating that the input from the second pathway is necessary to protect practiced patterns from being overwritten. Thus, interpreting the two input pathways as cortical and thalamic inputs to striatum, the model accounts for the experimental observations in rodents that skilled behaviors and neural representations of kinematic variables are robust to the loss of cortical inputs[29–31], but that such behaviors are not robust to the loss of thalamic input[28].

In summary, when an input pathway with fast supervised learning and another with slow Hebbian learning together drive a downstream population, the mechanisms of input alignment and control transfer enable the second input to gradually take over the role of the first in driving the downstream activity. For highly practiced patterns, this enables the second input to produce the correct downstream activity even if the first input is removed completely.

**Two-pathway model with reinforcement learning.** In the preceding sections we showed that a model combining fast learning in one pathway with slow Hebbian learning in a second pathway leads to the selective retention of patterns that are repeated multiple times, preventing these patterns from being overwritten by subsequent learning. We further showed that the mechanisms by which this effect occurs are input alignment, which causes the input currents from the two pathways to become aligned with one another, and control transfer, which causes the second pathway to become increasingly responsible for driving the downstream population as a pattern is repeated. In order to make the theory analytically tractable for calculations, we have so far considered an idealized scenario in which learning in the first pathway follows a supervised learning rule. A more biologically plausible scenario for learning at corticostriatal synapses, however, is reinforcement learning (RL), which requires stochastic updates to synaptic weights over many trials in order to maximize a scalar reward[14]. At first glance, the problems appear similar due to mathematical equivalence of maximizing a reward function in RL versus minimizing a loss function in supervised learning. However, the optimization is more daunting in the case of RL since it presumes less knowledge than supervised learning, making use only of a scalar reward signal, without being told which of all the possible outputs is the correct one. In this section, we show that the same mechanisms of input alignment and control transfer are realized in a version of the two-pathway model with RL rather than supervised learning in the first pathway. We further show that this version of the model provides insights into mechanisms underlying habit formation.

In order to describe RL in the first pathway, we assume that the readout neurons are activated stochastically, such that $z_i = +1$ with probability $\sigma(m_i + h_i)$ and $z_i = -1$ otherwise. Here $m_i = \sum_{j=1}^{N_x} W_{ij} x_j$ and $h_i = \sum_{j=1}^{N_y} V_{ij} y_j$ are the input currents from the two pathways introduced in the preceding section, and $\sigma(\lambda) = 1/(1 + e^{-\lambda})$ is the logistic sigmoid function. The goal of RL is to use trial-and-error learning in order to maximize a scalar reward, which we take to be the alignment between the readout activity

and a randomly selected target pattern, so that $R = \mathbf{z} \cdot \hat{\mathbf{z}} / \sqrt{N_z}$. The REINFORCE policy-gradient algorithm[48] (with reward computed relative to a baseline value[49]) provides a way of updating the weights in order to maximize the reward:

$$\Delta W_{ij} = \frac{\eta}{N_x}(R - \bar{R}) z_i \sigma(-z_i[m_i + h_i]) x_j, \qquad (5)$$

where $\eta$ is a learning rate, and $\bar{R}$ is a baseline reward expectation. The prefactor $R - \bar{R}$ in this learning rule is the reward prediction error, commonly identified with dopaminergic inputs to striatum[50].

It is instructive to compare Eq. (5) with the supervised update rule, according to which updates should be made along the direction $\Delta W_{ij} \propto \hat{z}_i x_j$ (Fig. 1c). Because the update in Eq. (5) is proportional to $z_i$ rather than to $\hat{z}_i$, the update may not be along the ideal direction in any given trial. However, since $R - \bar{R}$ will tend to be larger in trials for which $z_i = \hat{z}_i$ than in those for which $z_i = -\hat{z}_i$, we can see that the weights will be updated in the correct direction at least on average over many repetitions.

This suggests that RL should approximate the idealized case of supervised learning, but with learning of a single pattern taking place over many repetitions rather than just one. In turn, in order for Hebbian learning in the second pathway to reinforce the correct output pattern, learning in the second pathway must remain much slower than learning in the first pathway. Figure 4a shows that, in this limit, the mechanisms of input alignment and control transfer are realized in the two-pathway model with RL, similar to the two-pathway model with supervised learning.

As in the case with supervised learning, these effects allow for the input from the first pathway to be removed after a sufficient amount of practice without affecting the performance (Fig. 4a, bottom). In contrast, a model with fast RL in one pathway and slow RL in the other does not exhibit these effects and cannot produce the correct output when the first pathway is removed (Fig. 4b). This is because, although the input currents from the two pathways add together to produce the correct output in this case (black trace in Fig. 4b, bottom), there is no mechanism encouraging them to align with one another. Together, these results, applied to the biological interpretation of the model, suggest that RL at corticostriatal synapses together with Hebbian learning at thalamostriatal synapses may together account for the diminished dependence of highly practiced behaviors on motor cortex, whereas RL at both cortico- and thalamostriatal synapses cannot account for this effect.

In addition to reproducing the mechanisms of input alignment and control transfer found in the supervised learning case, the use of RL in the two-pathway model also enables us to address the mechanism of habit formation in greater detail. In general, a habit can be defined as a response that persists even when it is no longer rewarded. To examine habit formation, we trained two models to produce a particular output pattern $\hat{\mathbf{z}}^\nu$ in response to a random input pattern, repeating the association $n_\nu$ times, after which the target was switched to a new pattern $\hat{\mathbf{z}}^{\nu+1}$ while the input remained the same. In the first model, which featured RL in the first pathway and no second pathway, the activity in the readout population exhibited decreasing overlap with the first target pattern and increasing overlap with the second target pattern after the switch (Fig. 4c, top). In a two-pathway model with RL in the first pathway and Hebbian learning in the second pathway, on the other hand, the activity of the readout population exhibited an increased tendency to become stuck in alignment with the first target pattern (Fig. 4c, bottom). Compared with the single-pathway model, the activity of the readout population in the two-pathway model took longer to align with the new pattern after the switch if $n_\nu$ was small, or failed to align with it altogether

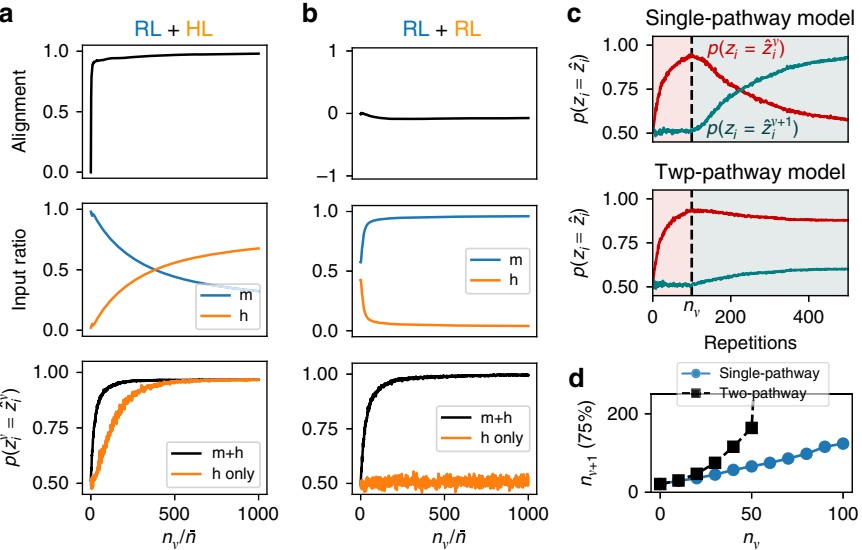

**Fig. 4 The two-pathway model with reinforcement learning (RL) and Hebbian learning (HL) accounts for input alignment, control transfer, and habit formation. a** A model trained on a single pattern with RL in the fast-learning pathway exhibits input alignment (top) and control transfer (middle), as in the version with supervised learning. After sufficient training, the correct output can still be produced if the input from the first pathway is removed (bottom). **b** Input alignment and control transfer do not occur in a model with fast RL in the first pathway and slow RL in the second. **c** A single-pathway model with RL only (top) or a two-pathway model with fast RL and slow HL (bottom) is trained for $n_\nu$ repetitions to map its input onto target output $\hat{\mathbf{z}}^\nu$, then is trained to map the same input onto a new target output $\hat{\mathbf{z}}^{\nu+1}$. **d** The number of repetitions necessary to produce the second target output pattern with 75% accuracy in the two models from panel (**c**). For all panels, the parameters used were $N_x = N_y = 1000$, $N_z = 10$, with learning rates $(\eta_1, \eta_2) = (1, 0.01)$ in (**b**), and learning rates $(\eta, \beta) = (1, 0.01)$ in other panels.

if $n_\nu$ was large (Fig. 4d). This result shows that Hebbian learning in the second pathway tends to promote habit formation, with the habit becoming more persistent as the initial stimulus-response association is repeated more times.

**Implementation in the sensorimotor brain circuit: automatization of behavior through practice.** In the previous sections, we showed that, for a single-layer neural network in which two input pathways operate with different types of learning, fast error-driven learning in the first pathway causes the downstream neurons to produce the correct output, while slow Hebbian learning in the second pathway reinforces the association between the input to the network and the correct output. After many repetitions, this scheme enables the second input pathway to assume control for driving the downstream layer. In this section, we develop the experimental interpretation of the model in greater detail, using it to address existing experimental results and to formulate predictions for future experiments.

In our neurobiological interpretation of the two-pathway model, we identify the readout population as sensorimotor striatum, the input population with fast supervised or reinforcement learning as motor cortex, and the input population with slow Hebbian learning as thalamus. Taken together, these identifications suggest a picture in which information about state, including sensory feedback, cues, and context, is represented in cortex and thalamus, and in which plasticity at the synapses descending from these structures to striatum allows for state information to be mapped onto actions that will tend to minimize errors and maximize rewards (Fig. 5a). The mechanisms and roles of learning in the two pathways are distinct, with fast changes to corticostriatal synapses serving to learn and produce the correct behavior, while slower changes to thalamostriatal synapses serve to associate a given state with a behavioral output, with this association becoming stronger and more persistent the more that it is practiced. As described in the previous section, the transfer of control from corticostriatal to thalamostriatal inputs would suggest that motor cortex should be necessary for the initial learning of a behavior, but not for the production of the behavior after a sufficient amount of practice. This is consistent with lesion and inactivation studies in rodents[29,31], as well as with studies of learned motor behaviors in humans suggesting that motor cortex becomes less involved in driving behaviors with practice[32,33].

With the neurobiological implementation proposed above, the two-pathway model also leads to a number of testable experimental predictions.

First and most obviously, the theory predicts that cortico- vs. thalamostriatal synapses should be modified by different types of plasticity (fast and reward-driven vs. slow and associative, respectively). As shown in Fig. 4, our theory suggests that existing experimental results are consistent with RL at corticostriatal synapses and Hebbian learning at thalamostriatal synapses, but inconsistent with RL at both sets of inputs. Because both types of synapses are glutamatergic, they have typically not been studied separately or even distinguished in past experiments[51]. Potentially relevant differences between these types of synapses are known to exist, however. For example, cortical inputs to striatum are more likely to terminate on dendritic spines, while thalamic inputs from the parafascicular and centromedian regions of thalamus are more likely to terminate on dendritic shafts[22,52]. These two types of synapses have also been shown to differentially express mu-opioid receptors[53] in mice, although this was in a different subregion of striatum (dorsomedial) than the one considered here (dorsolateral), and it is additionally unclear what the precise role of these receptors in synaptic plasticity might be. Future experiments, either in vivo or in slice preparations, could test whether these or other differences between cortico- and thalamostriatal synapses might underlie different ways in which they are modified during learning.

Second, because motor cortex plays less of a role in driving striatum as behaviors become highly practiced, the model predicts

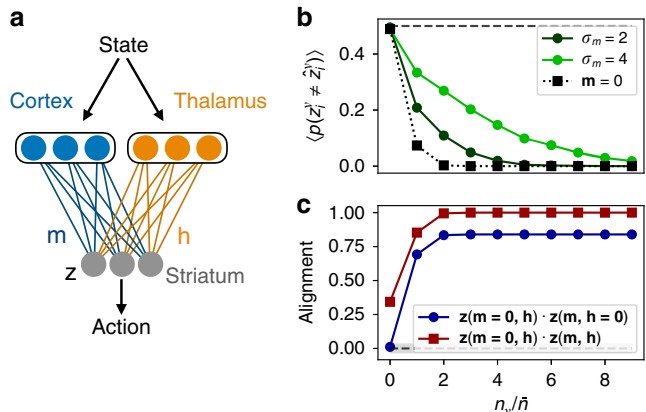

**Fig. 5 Two-pathway implementation in the sensorimotor brain circuit and predicted responses to perturbations. a** In the proposed neurobiological implementation of the two-pathway model, motor cortex and thalamus provide inputs to the sensorimotor region of striatum. **b** The population-averaged error rate in the readout population in the presence of noise perturbations to $\mathbf{m}$ (i.e., $m_i \rightarrow m_i + \sigma_m \xi_i$, where $\sigma_m$ is the noise amplitude and $\xi_i \sim \mathcal{N}(0, 1)$ is a Gaussian random variable; green lines) or upon removing the input $\mathbf{m}$ (dotted line) for a single pattern repeated $n_\nu$ times during training. **c** The normalized alignment between the population activity $\mathbf{z}$ with $\mathbf{m} = \mathbf{0}$ and with $\mathbf{h} = \mathbf{0}$ (blue line), and the normalized alignment between $\mathbf{z}$ with $\mathbf{m} = \mathbf{0}$ and with $\mathbf{m} \neq \mathbf{0}$. In (**b**) and (**c**), points are from simulations in a network with $\alpha = \beta = 1$ and $N_x = N_y = N_z = 1000$.

that lesioning, inactivating, or randomly perturbing activity in motor cortex during behavior should have a decreasingly small effect on a learned behavior the more that it has been trained (Fig. 5b). Furthermore, due to the fact that covert learning in the second pathway should occur slowly relative to learning in the first pathway, the learned behavior should become robust to motor-cortex inactivation only some time after the behavior has been mastered. This effect has in fact already been observed in very recent experiments using optogenetic inactivation throughout learning of a targeted forelimb task[31].

Third, online optogenetic inactivation could be used so that only motor cortex or only thalamus drives striatum in a given trial. The theory would predict that the striatal activities observed in these two cases would be correlated due to input alignment, with the degree of correlation increasing with the amount of training (Fig. 5c, blue line). Similarly, striatal population activity should be similar before and after cortical lesion or inactivation in a trained animal, with the degree of similarity increasing with the amount of training (Fig. 5c, red line). While recent work in rats has shown that movement kinematics can be decoded from sensorimotor striatum equally well before and after motor cortex lesion[30], future work will be needed to test whether the same pattern of neural activity is realized in striatum with versus without cortical input.

Finally, if thalamostriatal plasticity is blocked during training of a behavior, then the model predicts that the behavior will still be learned due to the fact that motor cortex is responsible for driving initial learning. Since input alignment cannot occur in the absence of thalamostriatal plasticity, however, the learned behavior will no longer be robust to cortical lesion or inactivation. This would be a challenging experiment to perform due to the fact that both thalamo- and corticostriatal synapses are glutamatergic, making selective manipulation difficult. If it could be done, however, it would provide strong evidence for our proposed neurobiological implementation of the two-pathway model.

**Simulated reaching task with the two-pathway network.** With the experimental interpretation from the preceding section in mind, we asked whether the basic mechanism illustrated in Fig. 2—namely, the enhanced retention of learned classifications that are practiced multiple times—could also be implemented in a neural network performing a more challenging sensorimotor task. We thus extended the two-pathway architecture to a neural network with one hidden layer and trained the network to perform a center-out reaching task (Fig. 6a, b).

In this task, the velocity of a cursor moving in two dimensions is controlled by a neural network, which receives as input the current location of the cursor together with a context input indicating which target to move toward. In each trial, the network was trained with supervised learning to follow a smooth, minimal-jerk trajectory to the specified target, with each trial having a duration of 10 timesteps. As before, the inputs were divided into two pathways, with supervised learning in one pathway and Hebbian learning in the other.

We trained a network to reach to target 1 in the first block of trials, then to target 2 in a second block, and so on, with more trials in the first block than in the later blocks (Fig. 6). Separately, we followed the same procedure to train a second network in which the Hebbian learning in the second pathway was inactivated by setting the learning rate to zero. Upon testing both networks, we found that recall of the first target, which had undergone extra repetitions, was significantly better in the network with Hebbian learning than in the network without Hebbian learning (Fig. 6d). This result shows that, in this trained network model as in the single-neuron case, Hebbian learning enables practiced behaviors to be better retained, protecting them from being overwritten as new behaviors are learned.

## Discussion

In this work, we have developed a bottom-up, mechanistic theory of fast and slow learning in parallel descending pathways, applying this theory to obtain insight into the possible roles of cortical and thalamic inputs to the basal ganglia in the acquisition and retention of learned behaviors through practice. While two-pathway architectures have been proposed previously as a means of addressing multistage learning in mammalian[54,55] and songbird[56] motor circuits, those models featured different learning mechanisms and were applied to different anatomical structures than the two-pathway model studied here. In particular, our theory proposes two simultaneously occurring learning processes: a fast process that minimizes errors in the cortical pathway, paired with a slow process that strengthens associations in the thalamic pathway. From a computational perspective, this work provides a quantitative theoretical framework in which the effects of learning, forgetting, and practice can be analyzed for the simplest possible case, namely a single neuron classifying random input patterns. From a cognitive perspective, the work provides an explanation for why repeated practice can continue to have effects even after a memory or behavior has been learned to be performed with zero error. From a neurobiological perspective, this work provides a framework for understanding the automatization of learned behavior and the formation of habits, proposing how these processes might be implemented in neural circuits.

In the Introduction, we described the multiple effects of practice: improved performance, habit formation, and decrease in cognitive effort[3]. The two-pathway model—and its proposed implementation in the brain's sensorimotor circuitry in particular—provides a unified account of these effects. First, fast learning in the corticostriatal pathway provides a mechanism for improved performance, with the greatest gains coming in the early stages of

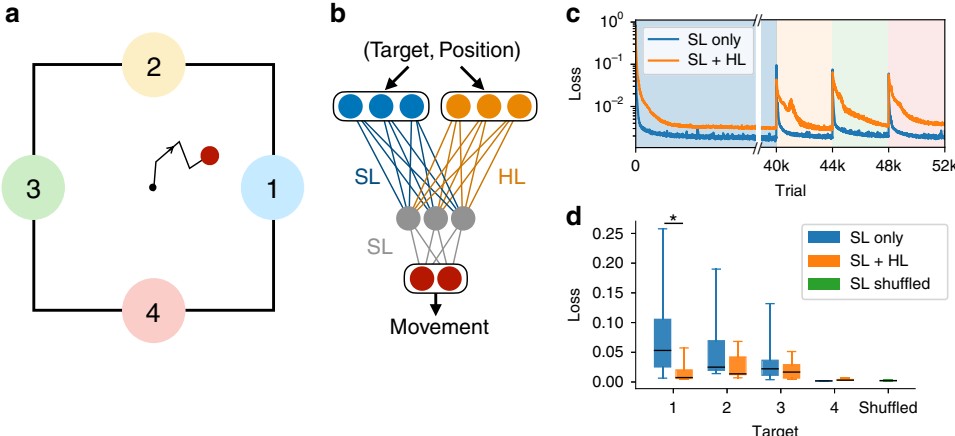

**Fig. 6 A two-pathway neural network trained to perform a reaching task exhibits enhanced retention of highly practiced behaviors. a** In this task, the agent, given one of four cues, is trained to perform a center-out reach over 10 timesteps to a particular target. **b** The neural network trained to perform the task receives a target-specific cue and the current position as input. The network contains one pathway learned with supervised learning (SL) and another with Hebbian learning (HL). The output is a two-dimensional vector determining the movement velocity in the next timestep. **c** The loss (mean squared error) on each of the four targets in sequence during training, with a longer training period for the first target, and with Hebbian learning either active (SL+HL) or inactive (SL only). **d** The loss (mean ± SEM) during testing on each of the four targets, after training is complete. The green bar shows a control network in which targets were randomly interleaved during training (*$p = 0.01$, two-sample two-sided $t$ test, $n = 21$ randomly initialized networks; boxes and whiskers denote 25th and 75th percentiles and 5th and 95th percentiles, respectively).

practice. Second, the two-pathway model provides a mechanistic description of habit formation, in which behaviors are produced consistently in response to particular stimuli, even when the behavior is no longer rewarded. This is because information about rewards and errors is encoded in the updates to corticostriatal synapses, which play less and less of a role in driving downstream activity the more that a behavior is practiced. Going further, the model may even provide a description of addiction and compulsion, in which behaviors persist even if they lead to negative outcomes. Finally, if we assume that cognitive control of behavior comes primarily from cortex, the two-pathway model provides a mechanistic description of behavioral automatization. As a behavior is repeatedly practiced, control is transferred to subcortical circuits, so that, without negatively impacting the behavior, the cortex becomes free to do other things even as the behavior is being performed (cf. Fig. 5b).

While the model that we have developed was constructed to address procedural learning of motor behaviors, the results shown in Fig. 2c–e suggest that it may also lead to insights for declarative memory. Not all of the results obtained from the two-pathway model were in agreement with published results from experiments on human memory, however. The fact that memory recall performance decays roughly as a power law has been relatively well established experimentally[46]. However, the forgetting curves in the two-pathway model are better described by exponential decay than by power laws (Supplementary Fig. 5). In addition, experimental study of spaced-repetition effects in human memory has shown that there exists an optimal interval between presentations during training[57], such that the interval between the two training presentations should match that between the second presentation and the time of testing. In the two-pathway model, however, we found that shorter training intervals always lead to better testing performance, regardless of the testing interval (Supplementary Fig. 6).

Of course, it is not surprising that a simple single-neuron model fails to capture the full range of experimental results on human memory. Indeed, it is not clear that our model should be expected to accurately describe declarative memory at all, since a great deal of research in neuroscience and psychology suggests that the hippocampal system is a crucial hub for the formation of

such memories, whereas the basal ganglia are more important for the formation of procedural memories including motor skills[58]. While it may be possible that ideas from the two-pathway model that we have developed could be applied to the hippocampal system, perhaps by including additional features such as complex synapses to obtain power-law memory decay[40,59] or higher-level cognitive strategies such as chunking to store information more efficiently[46], it is also possible that a theory of declarative memory will require entirely different architectures and learning rules than the ones that we have explored here.

In addition to exploring possible connections to declarative memory, future work should also fully extend the model from supervised learning to reinforcement learning, which is more difficult mathematically but more plausible biologically as a model of reward-based learning at corticostriatal synapses[14]. Reinforcement learning is also more plausible from a behavioral perspective, since it allows for performance to improve gradually by trial and error, leading to a richer description of learned motor behaviors than a single-step supervised learning rule can provide. While steps in this direction were taken in Fig. 4, which shows that the mechanisms of input alignment and control transfer are realized in a circuit trained with reinforcement learning, extending the theoretical framework underlying the two-pathway model to the case of reinforcement learning is an important direction for future work.

Finally, in this work, we have focused on sensorimotor striatum and its inputs. The thalamocortico-basal ganglia circuits that are involved in higher-level cognitive functions and that operate in parallel with the sensorimotor circuit could provide alternative interpretations of the model that we have presented[12,60]. More broadly, this two-pathway motif might be found in other brain areas and could in part explain the large degree of apparent redundancy found throughout the neural circuitry of the brain.

## Methods
Parameters and equations for the simulations in Figs. 1–5, as well as Supplementary Figs. 2 and 3, are provided in the main text. For Fig. 4, the expected reward for trial $n$ was computed as the low-pass filtered reward $\bar{R}^{n+1} = (1 - 1/\tau_R)\bar{R}^n + R^n/\tau_R$, where $R^n$ is the reward from trial $n$ and $\tau_R = 10$. Other parameters for producing Fig. 4 are provided in the main text. In Figs. 1–5,

results were averaged over $n = 1000$ trained networks, and error bars are comparable to or smaller than the size of the plotted points.

For the simulations in Fig. 6, $n = 21$ networks were trained for each of the conditions shown (SL only, SL+HL, and SL shuffled), where each network had $N_x = N_y = 50$, $N_z = 10$, and a 2-dimensional linear readout. The readout controlled the velocity of a cursor, which began at position $\mathbf{r} = \mathbf{0}$ and moved to one of the four target positions: $(1,0)$, $(0,1)$, $(-1,0)$, $(0,-1)$. The target velocity in each trial was given by

$$\hat{\mathbf{u}}(t) = \mathbf{r}^i \left[ 30(t/T)^2 - 60(t/T)^3 + 30(t/T)^4 \right],$$

where $t$ is the timestep, $T = 10$ is the number of timesteps in each trial, and $\mathbf{r}^i$ denotes one of the four target positions. This bell-shaped velocity profile describes a minimum-jerk trajectory from the initial to the target point. Both input layer populations receive fixed random projections of the network readout from the previous timestep, as well as a tonic random input associated with one of the four targets.

During training, the mean squared difference between the network output and $\hat{\mathbf{u}}(t)$ was minimized by gradient descent on the weights $\mathbf{W}$ (labeled SL in Fig. 6b), with learning rate $\eta_{SL} = 0.001$ (found by grid search). As a supervised learning algorithm that relies on backpropagation, gradient descent is not strictly biologically plausible. Because we are interested in investigating the effects of unsupervised learning in the second pathway, however, we have no reason to expect that our conclusions would strongly depend on the learning rule employed in the first pathway. The weights $\mathbf{V}$ (labeled HL in Fig. 6b) were either not trained (in the cases SL only and SL shuffled) or trained with the Hebbian update rule $\Delta V_{ij} = \eta_{HL}[-V_{ij} + z_i(t)y_j(t)]$, where $\eta_{HL} = 10^{-6}$ was found by grid search. The networks were sequentially trained on each target, as shown in Fig. 6c.

**Reporting summary**. Further information on research design is available in the Nature Research Reporting Summary linked to this article.

## Data availability

Python code for analyzing the simulation data from this work is available online at https://github.com/murray-lab/two-pathway-learning.

## Code availability

Python code for producing the simulation data from this work is available online at https://github.com/murray-lab/two-pathway-learning.

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

## Acknowledgements
The authors are grateful to B.P. Ölveczky, members of the Ölveczky lab, and L.F. Abbott for helpful discussions and feedback. Support for this work was provided by the National Science Foundation NeuroNex program (DBI-1707398), the National Institutes of Health (R01 NS105349, DP5 OD019897, U19 NS104649, and K99 NS114194), the Leon Levy Foundation, and the Gatsby Charitable Foundation.

## Author contributions
J.M.M. and G.S.E. performed the research and wrote the manuscript.

## Competing interests
The authors declare no competing interests.
