## [Peer Review File · Nature Communications]

Reviewers' Comments:

Reviewer #1:

Remarks to the Author:

This manuscript details a model of two different forms of synaptic plasticity onto a single neuron, one fast (one-shot perceptron learning) and one slow (Hebbian-style). The authors derive results on capacity and memory for the fast-alone and combined models, and show three key results: (1) recall above chance is maintained over a long history for sequentially-learned patterns; (2) in the combined model, "practice" (repetitions of a pattern) leads to almost perfect recall over a long history; (3) "practice" aligns the weights between the fast and slow input pathways - this covert learning creates redundancy between the two inputs, allowing either to give the correct output. They propose that the fast and slow forms of plasticity map onto the cortical and thalamic inputs to striatal medium spiny neurons, and in so doing outline how this mapping can explain key results on the role of cortex and basal ganglia in learning, in particular lesions of motor cortex.

This is a thought-provoking theory, and an original one to my knowledge. I especially appreciated the authors' careful separation of the learning models (Figs 2-3) from their proposed mapping - the results of Figs 2-3 stand regardless of the mapping. The paper is well-written, especially in its step-wise approach to introducing the models and their key results. There are a handful of places where clarity is lacking - and as those tend to be in key results I concentrate my comments there

Main comments:

(1) Why does the dual pathway model work? The simulation (Fig 2) show nicely that the two pathway model works - repeating a pattern means its correct output can be recalled near-perfectly after long intervals - but the main text lacks an explanation of why. The intuitive explanation seems to be that the Hebbian learning is sufficiently slow to only create meaningful synaptic updates from repeated patterns. So that pathway is in effect only contributing to the recall of the N repeated patterns. Presumably this means that if one includes many more repeated patterns (e.g. 5N), then one recovers a forgetting curve for the practiced patterns too - perhaps the authors should show us a forgetting curve for the slow pathway too (with sufficient repeated patterns) to get a sense of its capacity

(2) Input alignment and control transfer. This crucial section at the top of page 9 - its results underpin most of the proposed mapping to experimental results - was unclear to me. Two things in particular:
- The form of update equation. For the "fast" pathway, this is given as $m + \Delta m = z - h$; but "h" is the input current from the "slow" pathway. I could not work out where this cross-term came from - a typo?

- alignment: an explanation of why the rotation of h onto m is apparently inevitable would be good here. I understand the following, but it would be good to see it spelt out: because x and y are random vectors, so m and h are initially (approximately) orthogonal; thus for $z = \text{sgn}(m+h)$ to give the correct answer, so m and h must be moved closer than orthogonality; and as m is fixed once z is correct, so only h is rotating. Finally, the alignment reported seems extremely high (Fig 3c) - why to this level, rather than simply less-than-orthogonal?

Minor comments:

(1) pg 4. The statement "because the number of synapses N_x appears only through the parameter $[\gamma]$, the number of patterns that can be learned... is proportional to N_x ". I can't see from the form of Eq 26 and 28 that $P(z=1)$ is indeed strictly proportional to N_x - indeed the value of the ratio γ/N_x is itself not a constant. Clearly the memory capacity monotonically scales with N_x , but proportional?

(2) The reason for the form of Hebbian learning (Eq 3) is unclear. In particular: n_{μ} and $n_{\text{overscore}}$. " n_{μ} " is defined as "the number of times behaviour μ is repeated" - where one assumes "pattern" is meant. But if this the number of repeats of the pattern, then neither this number nor $n_{\text{overscore}}$ is accessible to a biological learning rule - for the number of patterns and future number of repeats of a pattern is unknown. These though are clearly just scaling constants for the two update terms in Eq 3. Could the authors comment on how they would be realised biologically?

(3) It was unclear what the reaching task (Fig 5) was demonstrating. The results show that the "SL only" models learns faster and better than the dual pathway model. Two things here:

- The sole advantage of the dual pathway model was in marginally better recall of the 1st trained target (Fig 5d). The main text claims "performance for target 1 was worse than that for the other targets... ($p=0.011$)". But Fig 5d implies only a statistical test between the two models on target 1, not between targets as claimed. Correct the claim, give the form of statistical test, and explain the asterisk

- It is unclear why the arm-reach model is in the main text, but the reinforcement learning model is relegated to the Supplementary material. Ultimately it is course up to the authors how they want to emphasise their material. But given the main claims of relevance to the cortico-striatal pathway, a convincing demonstration of a RL model in place of the perceptron rule seems relevant. That said, the RL rule used (Eq 61) is not strictly a synaptic learning rule either, containing only a learning rate, an error term (dopamine, in principle), and a noise term.

(4) The Discussion makes some strong claims in the paragraph starting "In the Introduction..." (pg 14). Two I did not entirely agree with:

- (i) that the model "provides a mechanistic description of habit formation". This is something of a stretch, as the model presented has no concept of rewards or errors, so no way of operationally defining "habits". For example, the crucial habitual behaviour test of devaluation cannot be done in the model. Certainly this model points the way to an improved theory of habit formation, but seems a bit strong to claim it's a mechanistic description.
- (ii) "with the amount of conscious effort devoted to a behaviour decreasing with increasing amounts of practice, as control of behaviour is transferred from a cortical to a subcortical circuit". Unclear here how the authors are defining "conscious effort", nor why that would be any less for a subcortical circuit than a cortical one.

(5) Suggested text changes

- Figure 1, panel c: explain the lines in the legend, not just the main text
- Eq 4: why the shift from " μ " to mean pattern to " v "?
- Figure 4: define "noise perturbations"
- pg 11 "Second, because motor cortex plays less of a role in driving striatum...": perhaps make clear this is a prediction of the model
- Figure 5: explain error bars in legend, not Methods

Mark Humphries

Reviewer #2:

Remarks to the Author:

This work proposes a feedforward model with two-pathways – one pathway that learns with the

perceptron and one pathway that learns unsupervised with Hebbian learning. The perceptron is mapped to fast cortical learning and the Hebbian learning is mapped to subcortical learning and habitual learning. The learned associations (weights learnt from the perceptron) are slowly transferred to the synapses undergoing Hebbian learning. They propose that the reason for this transfer is what they call input alignment. They show that the error rate is decreased for patterns that are repeated multiple times. They have a match of the forgetting curve between their analytical calculations and their numerical simulations. They finally simulated their model on a reaching task with the addition of a hidden layer.

This paper is well written, the derivations and the calculations are solid, and their results are well reported. The mapping between their very simple system and several brain areas is a little bit risky as their model is of course not very constrained to the biological details – not the neurons, not the architecture, not the learning rules (do we have evidence for supervised learning in cortex and Hebbian learning subcortically?, we typically think of cortex as Hebbian and subcortical as RL). Although simple models are better than complicated models, of course, maybe the authors should be careful of how far they can push their analogies.

Below are a set of comments:

- 1- It would be nice to know how the two-pathway model would behave under correlations in the patterns.
- 2- Similarly, the Fig. 5 does not take into account the temporal aspect of the task, but it would be great to know how the model would deal with that.
- 3- I think it would be good to know the behavior of the model for a regression instead of a classification (analogue perceptron). There, there are probably non-trivial results if the perceptron weights give an output with a remaining error.
- 4- The authors should compare their model to a two-pathway model with both pathways learning with the perceptron (so in that sense, it is similar to figure 1), but where the two pathways undergo two different learning rates, matching the learning rates of their original model.
- 5- I am not sure if I got the correct, but are the simulations done with the Hebbian learning term containing \hat{z} or real z ? They should do with real z of course, even if the analytics are done with \hat{z} .
- 6- The code should be given to the reviewers and released after publication.
- 7- Fig 3d bottom, $m+h$ looks like a rather minor improvement.
- 8- Fig 5. What do you need a hidden layer? Is it not linearly separable?
- 9- Fig 5. How do you train your hidden weights? Backprop?
- 10- What if, for Fig. 5, the model would also have a Hebbian pathway from the hidden layer?

Reviewer #3:

Remarks to the Author:

In this paper, Murray & Escola introduce a concise, straightforward computational model of the neural learning rules underlying motor skill learning, via dual input streams (cortical and thalamic) to the sensorimotor striatum. The authors design the proposed model primarily to characterize an influential finding in animal models of motor skill learning, namely lesion experiments suggesting that motor cortex is necessary for the acquisition of motor skills but not their long term retention (Kawai et al., 2015). Moreover, the model is designed to capture recent findings (both in preprint form, references [27, 29]) that specify the cortex-independent operation of the striatum in motor skill expression after extended practice, and the importance of thalamic inputs to the striatum for this process. The model posits that the algorithmic underpinnings of the transfer process from cortex- to striatum-mediated motor skill learning over practice reflects, respectively, supervised learning and Hebbian learning. The

model expresses several behaviors that match known findings about motor skill learning, its neural substrates, and the role of practice, and does so using very few computational assumptions. Moreover, the authors show that this model can be implemented in a simple neural network to control a plant, where the aforementioned learning characteristics can be observed.

Overall, I think this manuscript represents a compelling bottom-up theoretical approach to understanding the multiple processes/neural substrates driving motor skill learning. The techniques are thorough and (mostly) convincing, and the general idea of two qualitatively distinct learning rules driving, respectively, the corticostriatal and thalamostriatal components of motor skill learning, is a nice concept. I do, however, think there are some important gaps in the analysis, background, and interpretation that should be addressed to strengthen the manuscript. In general, several key points are brought up as future directions that appear to me to be somewhat fundamental.

Major Issues

The manuscript often discusses "habits" but, while I may be missing something, I do not see how the model or results can be said to reflect habitual behavior. The authors rightly define habits in the Introduction as a behavior showing an attenuation in its sensitivity to reward. I don't see how the forgetting curves, virtual lesions, or alignment results relate to habits. I think the authors could either a) remove the discussion of habit save some speculation in the Discussion, or b) do a more explicit model-based exploration of this idea, perhaps by expanding the fundamental role of RL in skill learning. This leads to the next point:

The discussion of a putative role of RL in the model (Figure S6) feels underdeveloped. First, what was the reward function and how was it implemented? Were variables like eligibility traces implemented? Why exactly did having RL in the thalamostriatal pathway fail? I think there's an interesting analysis in here but in its current form it is hard for the reader to understand or validate it.

While I appreciate greatly the discussion of the model's limitations on page 14, the main two fundamental discrepancies between the model output and the psychological literature seem problematic. First, as the authors show in some detail in Figure S4, their forgetting functions do not display the correct qualitative shape based on well-known empirical data. Is this a fundamental consequence of their model specification? Are there any dof in the parameterization that could remedy this flaw? Moreover, the model does not capture the spaced-versus-massed practice effect, and seems to actually suggest the opposite. I understand it could require too much additional experimentation (and could be open to a rebuttal that says as much), but I wonder if a "consolidation" process (within the cortical component rather than via a transfer between components) could be worked into the model.

This may be a somewhat vague concern, but I wonder if the authors could present any extant empirical work that lays the "biological plausibility" foundation for the model. That is, there could be a broader discussion of any neurophysiological evidence that corticostriatal synapses are more likely to undergo supervised learning and, conversely, thalamostriatal synapses are more likely to undergo Hebbian learning. I understand that this is a prediction of the model, but if that evidence exists it would be useful in the manuscript.

Minor Issues

Various items in Figure 1c should be defined in the legend (e.g., the red lines).

The sentences at the bottom of the second-to-last paragraph and the start of the last paragraph on page 14 are redundant.

Line numbers should be added

Reviewer #1 (Remarks to the Author):

This manuscript details a model of two different forms of synaptic plasticity onto a single neuron, one fast (one-shot perceptron learning) and one slow (Hebbian-style). The authors derive results on capacity and memory for the fast-alone and combined models, and show three key results: (1) recall above chance is maintained over a long history for sequentially-learned patterns; (2) in the combined model, “practice” (repetitions of a pattern) leads to almost perfect recall over a long history; (3) “practice” aligns the weights between the fast and slow input pathways - this covert learning creates redundancy between the two inputs, allowing either to give the correct output. They propose that the fast and slow forms of plasticity map onto the cortical and thalamic inputs to striatal medium spiny neurons, and in so doing outline how this mapping can explain key results on the role of cortex and basal ganglia in learning, in particular lesions of motor cortex.

This is a thought-provoking theory, and an original one to my knowledge. I especially appreciated the authors’ careful separation of the learning models (Figs 2-3) from their proposed mapping - the results of Figs 2-3 stand regardless of the mapping. The paper is well-written, especially in its step-wise approach to introducing the models and their key results. There are a handful of places where clarity is lacking - and as those tend to be in key results I concentrate my comments there

Main comments:

(1) Why does the dual pathway model work? The simulation (Fig 2) show nicely that the two pathway model works - repeating a pattern means its correct output can be recalled near-perfectly after long intervals - but the main text lacks an explanation of why. The intuitive explanation seems to be that the Hebbian learning is sufficiently slow to only create meaningful synaptic updates from repeated patterns. So that pathway is in effect only contributing to the recall of the N repeated patterns. Presumably this means that if one includes many more repeated patterns (e.g. $5N$), then one recovers a forgetting curve for the practiced patterns too - perhaps the authors should show us a forgetting curve for the slow pathway too (with sufficient repeated patterns) to get a sense of its capacity.

The mechanism underlying the enhanced recall of repeated patterns is what we term “control transfer” and is described in the section after the one that the reviewer is referring to. We agree that it is important to include at least a brief explanation and a promise of a fuller explanation later in the paper, so we have added the following text to the end of the paragraph discussing Fig. 2b: “The underlying reason for the selectively enhanced recall of practiced patterns...”

In order to address this point more fully, we have added Supplemental Figure 4, which illustrates the tradeoff between the enhancement of recall for patterns that are repeated during training versus the impairment of recall for the nonrepeated patterns. As the reviewer suggests, the conclusion that we draw from this figure is that the advantage of having repeated patterns starts to diminish as the number of repeated patterns becomes comparable to the number of inputs N .

(2) Input alignment and control transfer. This crucial section at the top of page 9 - its results underpin most of the proposed mapping to experimental results - was unclear to me. Two things in particular:

- The form of update equation. For the “fast” pathway, this is given as $m + \Delta m = z - h$; but “ h ” is the input current from the “slow” pathway. I could not work out where this cross-term came from - a typo?

This update equation is not a typo; the h comes from applying the update rule Eq. (1), which depends on the total input to the readout unit. We have added an intermediate step of its derivation in order to clarify where it comes from.

- alignment: an explanation of why the rotation of h onto m is apparently inevitable would be good here. I understand the following, but it would be good to see it spelt out: because x and y are random vectors, so m and h are initially (approximately) orthogonal; thus for $z = \text{sgn}(m+h)$ to give the correct answer, so m and h must be moved closer than orthogonality; and as m is fixed once z is correct, so only h is rotating. Finally, the alignment reported seems extremely high (Fig 3c) - why to this level, rather than simply less-than-orthogonal?

In order to clarify this point, we have added the following text to the

paragraph describing Figure 3c: “As described above, this increased alignment occurs because h becomes increasingly aligned with \hat{z} , and hence also with m , as the number of repetitions increases.”

Regarding the particular asymptotic value that the alignment assumes, we regret that we do not have a precise answer. We found that the analytical calculations of the curves in Figure 3c are surprisingly formidable. After some initial attempts at performing them, we decided that simulations alone would serve nearly as well for our purposes. An important point that should be clear from the arguments (including the new sentence) in the first paragraph of this section, though, is that we generically expect the alignment after many repetitions to be of order 1 (rather than vanishing as N becomes large). This is because both $m + \Delta m$ and $h + \Delta h$ lie in the plane spanned by \hat{z} and h .

Minor comments:

(1) pg 4. The statement “because the number of synapses N_x appears only through the parameter $[\gamma]$, the number of patterns that can be learned... is proportional to N_x ”. I can’t see from the form of Eq 26 and 28 that $P(z=1)$ is indeed strictly proportional to N_x - indeed the value of the ratio γ/N_x is itself not a constant. Clearly the memory capacity monotonically scales with N_x , but proportional?

We apologize that this point was not made more clearly and have rewritten the text to make it clearer. The point we intended to make is that the number of inputs N_x appears only through the ratio $(P-\nu)/N_x$, not that the probability in Eq. (2) is proportional to N_x . Hence, e.g., doubling the number of inputs will allow us to double the number of remembered patterns while maintaining a fixed error rate p .

(2) The reason for the form of Hebbian learning (Eq 3) is unclear. In particular: n_μ and $n_{\text{overscore}}$. “ n_μ ” is defined as “the number of times behaviour μ is repeated” - where one assumes “pattern” is meant. But if this the number of repeats of the pattern, then neither this number nor $n_{\text{overscore}}$ is accessible to a biological learning rule - for the number of patterns and future number of repeats of a pattern is unknown. These though are clearly just scaling constants for the two update terms in Eq 3.

Could the authors comment on how they would be realised biologically?

The reviewer is absolutely correct about this point. The mean number of patterns (as well as the number of inputs N_y) is not something that a real neuron would be able to calculate. These factors can clearly be absorbed into a redefinition of the learning rates, so in that sense they are arbitrary and unnecessary from a biological point of view. Writing Eq. (3) in the way that we have is a matter of mathematical convenience to make the results derived from this equation more transparent. We have added an explanation to the text below Eq. (3) to make this point clear (“While the prefactors...”).

(3) It was unclear what the reaching task (Fig 5) was demonstrating. The results show that the “SL only” models learns faster and better than the dual pathway model. Two things here:

- The sole advantage of the dual pathway model was in marginally better recall of the 1st trained target (Fig 5d). The main text claims “performance for target 1 was worse than that for the other targets... ($p=0.011$)”. But Fig 5d implies only a statistical test between the two models on target 1, not between targets as claimed. Correct the claim, give the form of statistical test, and explain the asterisk

Thanks to the reviewer for pointing out that our description of these results was unclear. We have rewritten the corresponding text to focus only on the statistically significant result indicated with the star in Figure 5d, and we have also included a note about the statistical test in the caption of Figure 5d.

- It is unclear why the arm-reach model is in the main text, but the reinforcement learning model is relegated to the Supplementary material. Ultimately it is course up to the authors how they want to emphasise their material. But given the main claims of relevance to the cortico-striatal pathway, a convincing demonstration of a RL model in place of the perceptron rule seems relevant. That said, the RL rule used (Eq 61) is not strictly a synaptic learning rule either, containing only a learning rate, an error term (dopamine, in principle), and a noise term.

We are grateful to the reviewer for encouraging us to put greater emphasis

on the role of reinforcement learning in our model. We have added a new section to the main text including a new figure showing that the mechanisms of input alignment and control transfer are realized in the two-pathway model with RL. This section also addresses the issue of habit formation, which the reviewer brought up in his next question.

(4) The Discussion makes some strong claims in the paragraph starting “In the Introduction...” (pg 14). Two I did not entirely agree with:

(i) that the model “provides a mechanistic description of habit formation”. This is something of a stretch, as the model presented has no concept of rewards or errors, so no way of operationally defining “habits”. For example, the crucial habitual behaviour test of devaluation cannot be done in the model. Certainly this model points the way to an improved theory of habit formation, but seems a bit strong to claim it’s a mechanistic description.

In response to the reviewers’ comments, we have added a new section on reinforcement learning to the main text. The new figure in that section explicitly addresses the issue of habit formation, showing that learning in the second pathway makes it more difficult for the circuit to learn a new output in response to a stimulus that it has previously associated with a different output. We believe that this will fully address the reviewer’s concern.

(ii) “with the amount of conscious effort devoted to a behaviour decreasing with increasing amounts of practice, as control of behaviour is transferred from a cortical to a subcortical circuit”. Unclear here how the authors are defining “conscious effort”, nor why that would be any less for a subcortical circuit than a cortical one.

We regret that this point was not made sufficiently clear in the original version of the manuscript, and we thank the reviewer for prompting us to improve it. We have replaced the sentence with the following:

“Finally, if we assume that cognitive control of behavior comes primarily from cortex, the two-pathway model provides a mechanistic description of behavioral automatization. As a behavior is repeatedly practiced, control is transferred to subcortical circuits, so that, without negatively impacting the

behavior, the cortex becomes free to do other things even as the behavior is being performed (cf. Fig. 5b).”

(5) Suggested text changes

- Figure 1, panel c: explain the lines in the legend, not just the main text

We have included a description of the two red lines from Figure 1c in the figure caption.

- Eq 4: why the shift from “mu” to mean pattern to “v”?

Throughout the text, we have used μ as a generic index for a pattern that is being trained, then ν as a generic index for a pattern that is being tested. Though reasonable opinions might differ on this, we felt that doing things this way would minimize potential confusion about whether a given equation applies during training or during testing, and we would prefer to keep the index notation as it currently stands.

- Figure 4: define “noise perturbations”

We have included a definition of “noise perturbations” in the caption of Figure 4b.

- pg 11 “Second, because motor cortex plays less of a role in driving striatum...”: perhaps make clear this is a prediction of the model

We have added “the model predicts that...” to the sentence that the reviewer mentioned.

- Figure 5: explain error bars in legend, not Methods

The error bars are now described in the figure caption.

Reviewer #2 (Remarks to the Author):

This work proposes a feedforward model with two-pathways – one pathway that learns with the perceptron and one pathway that learns unsupervised

with Hebbian learning. The perceptron is mapped to fast cortical learning and the Hebbian learning is mapped to subcortical learning and habitual learning. The learned associations (weights learnt from the perceptron) are slowly transferred to the synapses undergoing Hebbian learning. They propose that the reason for this transfer is what they call input alignment. They show that the error rate is decreased for patterns that are repeated multiple times. They have a match of the forgetting curve between their analytical calculations and their numerical simulations. They finally simulated their model on a reaching task with the addition of a hidden layer.

This paper is well written, the derivations and the calculations are solid, and their results are well reported. The mapping between their very simple system and several brain areas is a little bit risky as their model is of course not very constrained to the biological details – not the neurons, not the architecture, not the learning rules (do we have evidence for supervised learning in cortex and Hebbian learning subcortically?, we typically think of cortex as Hebbian and subcortical as RL). Although simple models are better than complicated models, of course, maybe the authors should be careful of how far they can push their analogies.

Regarding the last point made here by the reviewer, we agree with the reviewer that reinforcement learning is a much better candidate than supervised learning for describing corticostriatal learning in the brain. We regret that the original version of our manuscript didn't present the philosophy of our approach more clearly. Rather than arguing that SL is a better description than RL for how learning works in the brain, we mean to argue that SL is a reasonable approximation to RL. We provided some evidence for this in Supplemental Figure 6 in the original version, but, in response to the reviewers' comments, have updated it and moved it to the main text, where we have added a new section that more clearly lays out the way that RL fits into our theory.

The reviewer also points out that the learning rule we study is not sufficiently biologically motivated. As we mention in our paper and as the reviewer would no doubt agree, there is a good deal of evidence that some form of reinforcement learning occurs at corticostriatal synapses. In the initial version of our paper, we used supervised learning rather than RL at these synapses, thinking of it as an idealized and mathematically tractable

version of RL. We regret that we did not make this connection sufficiently explicit, however, and we have remedied this in the current version with a new section on the two-pathway theory with RL.

The other learning rule invoked by our model is Hebbian learning at thalamostriatal synapses. This is a new idea, and hence has not yet been experimentally proven or disproven. As we describe in our Introduction, it is motivated by circuit-level phenomena including lesion and inactivation results during learning in rodents. And as we describe in our Discussion, it has not to our knowledge (nor to the knowledge of experts we've asked) been tested by existing experiments. Although, as we show in Figure 4a-b, our theory provides evidence that existing experimental results are consistent with Hebbian learning but inconsistent with RL at thalamostriatal synapses. Our theory, based on the assumption of Hebbian learning, also generates additional predictions that will allow for this assumption to be tested by future experiments, as described in the final Results subsection of our paper.

Below are a set of comments:

1- It would be nice to know how the two-pathway model would behave under correlations in the patterns.

Please see our response to comment #3 below.

2- Similarly, the Fig. 5 does not take into account the temporal aspect of the task, but it would be great to know how the model would deal with that.

The meaning of this comment isn't clear to us. Precisely because of the temporal aspect of the task, the inputs are correlated across time steps, but it isn't clear whether this is the "temporal aspect" that the reviewer has in mind. If this point is important to the reviewer, we would be happy to try to address it given some further clarification.

3- I think it would be good to know the behavior of the model for a regression instead of a classification (analogue perceptron). There, there are probably non-trivial results if the perceptron weights give an output with a remaining error.

A main reason that we included the simulated reaching task shown in Figure 6 (formerly Figure 5) as part of our study was to show that the basic mechanisms that we are interested in describing (specifically, the enhanced retention of highly practiced patterns) can still emerge in a multilayer network solving a complex task, where this task exhibits temporal correlations between patterns (addressing comment 1 above) as well as nonbinary outputs (addressing this comment).

While we completely agree that it would be worthwhile to have a fully systematic understanding of how the model behaves with nonbinary neurons and input pattern correlations (comment #1 above), our view is that investigating this properly, i.e. extending the drift-diffusion calculations that form the core of our theory, ideally for the case of reinforcement learning in addition to supervised learning, would be a significant undertaking that is best left for a future paper. With the Editors' permission, we would prefer not to further develop our theory in this direction beyond the results already shown in Figure 6.

4- The authors should compare their model to a two-pathway model with both pathways learning with the perceptron (so in that sense, it is similar to figure 1), but where the two pathways undergo two different learning rates, matching the learning rates of their original model.

This is a great idea, but it would not be possible to do this meaningfully using the supervised learning rule that we used since the fast pathway reduces the output error to zero after a single iteration. As a consequence, there will be no further updates in either pathway since there is no error left to minimize.

This idea can, however, be applied to the case of reinforcement learning. In response to the reviewer's comment, we have done this in Figure 4b and shown that input alignment and control transfer do not occur in a two-pathway model with fast RL in one pathway and slow RL in the other.

5- I am not sure if I got the correct, but are the simulations done with the Hebbian learning term containing \hat{z} or real z ? They should do with real z of course, even if the analytics are done with \hat{z} .

The simulations were done with a Hebbian learning rule including z rather than \hat{z} . The learning rule is given explicitly in the Materials and Methods section.

6- The code should be given to the reviewers and released after publication.

We have submitted three Python notebooks which reproduce the main results from our paper for the reviewers to review. We will be happy to have this code published along with the paper.

7- Fig 3d bottom, $m+h$ looks like a rather minor improvement.

The main point of this figure is not the overall shift in the curve when the second pathway is included, but rather the fact that the highly practiced pattern is remembered perfectly even after the loss of the first pathway. We apologize for this point of confusion and have added a clarification (“for this pattern”) to the text discussing the figure in order to minimize confusion.

8- Fig 5. What do you need a hidden layer? Is it not linearly separable?

Our main motivation for including the hidden layer was driven by neuroanatomy rather than by computational considerations, since we think of the striatum as a hidden layer in a feedforward motor circuit that produces learned actions. We have checked that the network can be trained to a comparable level of performance when it does not include the second layer (data not shown), but, with the reviewers’ permission, we would prefer to keep the hidden layer in these simulations. In addition to closer correspondence with neuroanatomy, we believe that many readers will also be interested to see that the two-pathway motif can fruitfully be applied to multilayer architectures.

9- Fig 5. How do you train your hidden weights? Backprop?

Yes, the hidden weights are trained with gradient descent (i.e. backprop), as stated in the Methods section.

10- What if, for Fig. 5, the model would also have a Hebbian pathway from

the hidden layer?

We did not try this because we don't think that such an architecture is well-motivated by the biological circuit (i.e. we know of no evidence for separate pathways with different learning rules downstream from striatum).

Reviewer #3 (Remarks to the Author):

In this paper, Murray & Escola introduce a concise, straightforward computational model of the neural learning rules underlying motor skill learning, via dual input streams (cortical and thalamic) to the sensorimotor striatum. The authors design the proposed model primarily to characterize an influential finding in animal models of motor skill learning, namely lesion experiments suggesting that motor cortex is necessary for the acquisition of motor skills but not their long term retention (Kawai et al., 2015). Moreover, the model is designed to capture recent findings (both in preprint form, references [27, 29]) that specify the cortex-independent operation of the striatum in motor skill expression after extended practice, and the importance of thalamic inputs to the striatum for this process. The model posits that the algorithmic underpinnings of the transfer process from cortex- to striatum-mediated motor skill learning over practice reflects, respectively, supervised learning and Hebbian learning. The model expresses several behaviors that match known findings about motor skill learning, its neural substrates, and the role of practice, and does so using very few computational assumptions. Moreover, the authors show that this model can be implemented in a simple neural network to control a plant, where the aforementioned learning characteristics can be observed.

Overall, I think this manuscript represents a compelling bottom-up theoretical approach to understanding the multiple processes/neural substrates driving motor skill learning. The techniques are thorough and (mostly) convincing, and the general idea of two qualitatively distinct learning rules driving, respectively, the corticostriatal and thalamostriatal components of motor skill learning, is a nice concept. I do, however, think there are some important gaps in the analysis, background, and interpretation that should be addressed to strengthen the manuscript. In

general, several key points are brought up as future directions that appear to me to be somewhat fundamental.

Major Issues

The manuscript often discusses “habits” but, while I may be missing something, I do not see how the model or results can be said to reflect habitual behavior. The authors rightly define habits in the Introduction as a behavior showing an attenuation in its sensitivity to reward. I don’t see how the forgetting curves, virtual lesions, or alignment results relate to habits. I think the authors could either a) remove the discussion of habit save some speculation in the Discussion, or b) do a more explicit model-based exploration of this idea, perhaps by expanding the fundamental role of RL in skill learning. This leads to the next point:

The discussion of a putative role of RL in the model (Figure S6) feels underdeveloped. First, what was the reward function and how was it implemented? Were variables like eligibility traces implemented? Why exactly did having RL in the thalamostriatal pathway fail? I think there’s an interesting analysis in here but in it’s current form it is hard for the reader to understand or validate it.

In response to the reviewer’s comments (related also to those of Reviewer 1 above), we have included a new section on RL in the two-pathway model in the main text. This section addresses all of the questions that the reviewer brought up, specifically:

- *What was the reward function?* As we describe in the text, the reward function is the overlap between the activity of the readout population (“striatum”) and a target readout state.
- *Were variables like eligibility traces implemented?* Since there are no temporal delays in credit assignment, eligibility traces for the synaptic traces were not used, though a moving average of the reward over trials was used in order to compute a reward prediction error for learning, as described in the text.

- *Why exactly did having RL in the thalamostriatal pathway fail?* As we describe in the text, although the input currents from the two pathways add up to produce the correct output in the case of RL in both pathways, there is no mechanism encouraging them to line up with one another, so the control transfer phenomenon does not occur in this case.

We believe that this expansion of the RL results has significantly strengthened our paper, and we are grateful to the reviewer for encouraging us to develop our ideas and results in this direction.

While I appreciate greatly the discussion of the model's limitations on page 14, the main two fundamental discrepancies between the model output and the psychological literature seem problematic. First, as the authors show in some detail in Figure S4, their forgetting functions do not display the correct qualitative shape based on well-known empirical data. Is this a fundamental consequence of their model specification? Are there any dof in the parameterization that could remedy this flaw? Moreover, the model does not capture the spaced-versus-massed practice effect, and seems to actually suggest the opposite. I understand it could require too much additional experimentation (and could be open to a rebuttal that says as much), but I wonder if a "consolidation" process (within the cortical component rather than via a transfer between components) could be worked into the model.

As the reviewer is no doubt aware, a great deal of experimental work has pointed to a "two systems" framework for understanding memory, in which declarative/semantic/episodic memories depend mainly on the hippocampus, while nondeclarative/procedural/motor memories depend mainly on the basal ganglia. After some initial hopes that our model might serve as a grand unified theory of all memory, the results in Figs. S4-5 together with discussions with experimental colleagues in neuroscience and psychology have convinced us that our model is much more appropriate for describing nondeclarative rather than declarative memory. In addition to the discrepancies in Figs. S4-5, it is also unclear whether there is a mapping of our circuit architecture and learning rules onto the neuroanatomy of the hippocampus, where declarative memory is thought to take place. Thus, we view our model as being broadly consistent with the two-systems framework, describing one of these systems but not

necessarily the other.

The reviewer's questions about whether the model could be made more consistent with the experimental psychology results on power-law memory decay and spaced repetition is nevertheless a reasonable one. In the Discussion section we speculate on that multistate synapses (as in the cascade model of Stefano Fusi and colleagues, in which synaptic weights, rather than being represented by a single value, are represented with multiple values that influence one another and change on different timescales) might be fruitfully incorporated into our model to address this point. Because we don't see a clear mapping of our circuit onto the hippocampus for describing declarative memory, and because we aren't aware of results in the motor literature pointing to power law forgetting curves or optimal spaced repetition intervals, our preference is to not pursue these ideas further beyond some speculation in our Discussion section.

In response to the reviewer's comments and with the above points in mind, we have rewritten the paragraph in the Discussion mentioning Figs. S4-5 to more clearly delineate the realm of applicability of our theory.

This may be a somewhat vague concern, but I wonder if the authors could present any extant empirical work that lays the "biological plausibility" foundation for the model. That is, there could be a broader discussion of any neurophysiological evidence that corticostriatal synapses are more likely to undergo supervised learning and, conversely, thalamostriatal synapses are more likely to undergo Hebbian learning. I understand that this is a prediction of the model, but if that evidence exists it would be useful in the manuscript.

As we mention in the Introduction, the most likely candidate for the form of learning at corticostriatal synapses is generally believed to be reinforcement learning (RL). Our intention in constructing our model with supervised learning was not to claim that this is actually the form of learning occurring at corticostriatal synapses, but rather that supervised learning is a reasonable approximation to RL while also being more mathematically tractable. We regret that this point was not made more clearly in the initial version of our manuscript and have addressed the

relationship between supervised learning and RL more explicitly in the new RL section that we have added to the paper (essentially, RL can be thought of as a noisy version of supervised learning, following the same update direction in the space of synaptic weights in the limit of a very large number of very small updates for each pattern).

Regarding the thalamostriatal plasticity, we have consulted with some synaptic physiologists who work on striatum. Their view, which is in line with our own reading of the literature, has been that there is neither strong evidence for nor against the possibility of slow Hebbian plasticity at these synapses, as predicted by our theory, but that it is certainly within the realm of plausibility. In response to the reviewer's suggestion, we have added references to the limited experimental results that we know of on potentially relevant differences between cortico- and thalamostriatal synapses (In the first paragraph on experimental predictions: "Potentially relevant differences between these types of synapses...").

Minor Issues

Various items in Figure 1c should be defined in the legend (e.g., the red lines).

We have included more details about this figure in the caption.

The sentences at the bottom of the second-to-last paragraph and the start of the last paragraph on page 14 are redundant.

We have rewritten the first of these sentences to address the redundancy.

Line numbers should be added

We have added line numbers to facilitate review and revision.

Reviewers' Comments:

Reviewer #1:

Remarks to the Author:

The authors have done an excellent job of revising an already great paper. The new RL model work is an excellent addition, and makes the crucial point that there is little work comparing plasticity or dopamine modulation of cortico-striatal and thalamo-striatal synapses.

Minor comments:

(1) The effect of the number of repeated patterns - please check and clarify throughout where and how this is tested. Supplemental Figure 4 title claims to show the effect of the number of repeated patterns (n^*), but I cannot see it. The last line on page 7 (marked-up copy) says "In Supplemental Figure ??, we show that this remains true as long as the number of repeated patterns is much smaller than the total number of inputs $N_x + N_y$ ", but it was unclear which Figure this was referring to - if Supp Figure 4, then it does not?

(2) A couple of quibbles with the RL model discussion in sections 4 and 5.

(a) As the purpose of this model is to tie the model more closely to the dopamine-modulated cortico-striatal plasticity, I would quibble with the claim that $R-R(\text{baseline})$ is the reward prediction error as usually understood in the striatum to arise from the dopamine signal. As the definition of R here is entirely intrinsic to the model, and its maximum value is obtained from alignment of the output and target vectors, so we'd expect it to increase to an asymptote with repeats of the input pattern, which means the maximum RPE is after many repeats. But from a behavioural standpoint, we'd expect the largest $R-R(\text{baseline})$ to be on the first correct trial (i.e. the first time there is a "sufficient" alignment of the vectors), and so towards the start of the repeats. There is also the increasing $R(\text{baseline})$ in the Methods, which is both odd and, as far as I can tell, does not counter the maximum of $R-R(\text{baseline})$ rising to an asymptote (but defines the asymptote). Suggest a clarification here in the text, and perhaps mention of this issue in the discussion.

(b) pg 14 (of the marked-up copy): contrary to what is claimed here, thalamo-striatal synapses do target the spines of SPNs; to my knowledge, it is the synapses originating from Pf/CM thalamus that target the shaft. (See Smith et al (2009) The thalamostriatal systems: Anatomical and functional organization in normal and parkinsonian states, Brain Research Bulletin). Suggest a slight re-wording here.

(X) Suggested text changes

- pg 7 of the "mark-up" copy references to an unknown Supplemental Figure ("??")
- Supplemental Figure 4: the "text" and legend repeat the same information, but in different terminology, confusing this reader a little
- why is $R = z \cdot \hat{z} / \sqrt{N_z}$ rather than \hat{z} / N_z (and so in $[-1,1]$)?

Mark Humphries

Reviewer #2:

Remarks to the Author:

I don't have any other concerns. One little thing - I would encourage the authors to mention clearly that learning in their hidden layer is done by backprop and there that it is not biologically plausible (as it uses not local information).

Reviewer #3:

Remarks to the Author:

I am pleased with the authors' significant expansion of the RL section, and the more explicit connections to habit. I also think the caveats on memory dynamics and the MTL in the Discussion are an important addition.

Small comment: I think the equating of SL and RL by converting the loss function to a reward function should be emphasized more in the text as it could be jarring for some readers. Perhaps a few sentences describing this step and/or citing work that supports these flexible definitions could be helpful (e.g., Barto & Dietterich '04).

We have addressed the remaining reviewer comments in our point-by-point response below. We would like to once again reiterate our thanks to Prof. Humphries and the other two referees for their insightful comments and critiques.

REVIEWERS' COMMENTS

Reviewer #1 (Remarks to the Author):

The authors have done an excellent job of revising an already great paper. The new RL model work is an excellent addition, and makes the crucial point that there is little work comparing plasticity or dopamine modulation of cortico-striatal and thalamo-striatal synapses.

Minor comments:

(1) The effect of the number of repeated patterns - please check and clarify throughout where and how this is tested. Supplemental Figure 4 title claims to show the effect of the number of repeated patterns (n^*), but I cannot see it. The last line on page 7 (marked-up copy) says “In Supplemental Figure ??, we show that this remains true as long as the number of repeated patterns is much smaller than the total number of inputs $N_x + N_y$ ”, but it was unclear which Figure this was referring to - if Supp Figure 4, then it does not?

We apologize for the missing figure reference. The quoted text does indeed refer to Supplemental Figure 4, and this has been corrected. Panel b of this figure shows the dependence of the areas defined in panel a, which we take to be metrics of the memory performance, on n^* . In order to make it clearer what the quantities being plotted here are and how they relate to memory performance, we have added the following text to the description of Supplemental Figure 4: “In Supplemental Figure \ref{fig:reps}a, we define the green shaded area as a metric of how well the repeated patterns are retained, and we define the red shaded area as a metric of how well the non-repeated patterns are retained. (The area between the dashed and dotted lines provides a baseline in which no patterns are repeated multiple times.)”

(2) A couple of quibbles with the RL model discussion in sections 4 and 5.

(a) As the purpose of this model is to tie the model more closely to the dopamine-modulated cortico-striatal plasticity, I would quibble with the claim that $R-R(\text{baseline})$ is the reward prediction error as usually understood in the striatum to arise from the dopamine signal. As the definition of R here is entirely intrinsic to the model, and its maximum value is obtained from alignment of the output and target vectors, so we'd expect it to increase to an asymptote with repeats of the input pattern, which means the maximum RPE is after many repeats. But from a behavioural standpoint, we'd expect the largest $R-R(\text{baseline})$ to be on the first correct trial (i.e. the first time there is a “sufficient” alignment of the vectors), and so towards the start of the repeats. There is also the increasing $R(\text{baseline})$ in the Methods, which is both odd and, as far as I can tell, does not counter the maximum of $R-R(\text{baseline})$ rising to an asymptote (but defines

the asymptote). Suggest a clarification here in the text, and perhaps mention of this issue in the discussion.

While the reviewer's prediction accurately describes the trend of the *reward* over training, where this quantity increases monotonically, the *RPE*, due to the subtracted baseline, does not in fact behave in the same way. As one would expect from classic temporal difference learning theory, and in accord with the reviewer's expectations, the RPE takes its largest values in early trials and then decreases to zero after further training as the reward comes to match the expected reward. We have verified this in simulations (data not shown, but available upon request).

We believe that the root of the misunderstanding must come from a misreading of the definition of the baseline \bar{R} , which the reviewer mistakenly refers to as "increasing" and "odd". In fact it is neither of these things, but simply the low-pass filtered reward over recent trials, as conventionally used in reinforcement learning. We apologize if this was not clear and have fixed it by adding the clarification "low-pass filtered reward" to the definition of \bar{R} in the first paragraph of the Methods section.

(b) pg 14 (of the marked-up copy): contrary to what is claimed here, thalamo-striatal synapses do target the spines of SPNs; to my knowledge, it is the synapses originating from Pf/CM thalamus that target the shaft. (See Smith et al (2009) The thalamostriatal systems: Anatomical and functional organization in normal and parkinsonian states, Brain Research Bulletin). Suggest a slight re-wording here.

The reviewer is right to point out our overgeneralization. We have added the qualification "from the parafascicular and centromedian regions of thalamus" in order to avoid confusion. We have also added a reference to the paper cited by the reviewer.

(X) Suggested text changes

- pg 7 of the "mark-up" copy references to an unknown Supplemental Figure ("??")

This has been fixed.

- Supplemental Figure 4: the "text" and legend repeat the same information, but in different terminology, confusing this reader a little

We have made some minor adjustments and clarifications to the supplementary text describing Supplementary Figure 4. (See also our response to Comment 1 above.)

- why is $R = z \cdot \hat{z} / \sqrt{N_z}$ rather than $z \cdot \hat{z} / N_z$ (and so in $[-1, 1]$)?

Because the elements of z are randomly chosen as ± 1 , our definition gives an R that is, by the central limit theorem, of order 1 as N_z becomes large.

Mark Humphries

Reviewer #2 (Remarks to the Author):

I don't have any other concerns. One little thing - I would encourage the authors to mention clearly that learning in there hidden layer is done by backprop and there that it is not biologically plausible (as it uses not local information).

Thanks to the reviewer for this suggestion. In order to address this point, we have added the following clarification to the third paragraph of the Methods section: "As a supervised learning algorithm that relies on backpropagation, gradient descent is not strictly biologically plausible. Because we are interested in investigating the effects of unsupervised learning in the second pathway, however, we have no reason to expect that our conclusions would strongly depend on the learning rule employed in the first pathway."

Reviewer #3 (Remarks to the Author):

I am pleased with the authors' significant expansion of the RL section, and the more explicit connections to habit. I also think the caveats on memory dynamics and the MTL in the Discussion are an important addition.

Small comment: I think the equating of SL and RL by converting the loss function to a reward function should be emphasized more in the text as it could be jarring for some readers. Perhaps a few sentences describing this step and/or citing work that supports these flexible definitions could be helpful (e.g., Barto & Dietterich '04).

Thanks to the reviewer for this suggestion. In order to address this point, we have added the following text to the first paragraph in the section on RL: "At first glance, the problems appear similar due to mathematical equivalence of maximizing a reward function in RL versus minimizing a loss function in supervised learning. However, the optimization is more daunting in the case of RL since it presumes less knowledge than supervised learning, making use only of a scalar reward signal, without being told which of all the possible outputs is the correct one."